# Transparent EEG Analysis: Leveraging Autoencoders, Bi-LSTMs, and SHAP for Improved Neurodegenerative Diseases Detection

**DOI:** 10.3390/s25185690

**Published:** 2025-09-12

**Authors:** Badr Mouazen, Ahmed Bendaouia, Omaima Bellakhdar, Khaoula Laghdaf, Aya Ennair, El Hassan Abdelwahed, Giovanni De Marco

**Affiliations:** 1LINP2 Lab, Paris Nanterre University, UPL Paris, 92000 Nanterre, France; gdemarco@parisnanterre.fr; 2Institute for Advanced Manufacturing (IAM), University of Texas Rio Grande Valley, Edinburg, TX 78539, USA; ahmed.bendaouia@utrgv.edu; 3LISI Lab, Computer Science Department, Faculty of Sciences Semlalia, Cadi Ayyad University, Marrakech 40000, Morocco; o.bellakhdar6777@uca.ac.ma (O.B.); k.laghdaf2035@uca.ac.ma (K.L.); a.ennair6582@uca.ac.ma (A.E.); abdelwahed@uca.ac.ma (E.H.A.)

**Keywords:** EEG analysis, Alzheimer’s disease, frontotemporal dementia, deep learning, autoencoder, bidirectional LSTM, SHAP, explainable AI, neurodegenerative diseases, machine learning, signal processing, temporal analysis

## Abstract

**Highlights:**

Novel hybrid architecture: Combined autoencoders with bidirectional LSTM networks for enhanced EEG signal classification, achieving 98% accuracy in distinguishing AD, FTD, and healthy controls.Explainable AI integration: Implemented SHAP (SHapley Additive exPlanations) framework to enhance model transparency and identify entropy as the most influential feature for neurodegenerative disease detection.Optimal temporal segmentation: Demonstrated that 5-s EEG windows with 50% overlap provide the best balance between classification accuracy and computational efficiency.Comprehensive feature extraction: Utilized Power Spectral Density (PSD) analysis across standard frequency bands (Delta, Theta, Alpha, Beta, Gamma) following autoencoder-based dimensionality reduction.Superior performance validation: Outperformed traditional machine learning methods (KNN: 38%, SVM: 40%) and unidirectional LSTM (84%) with the proposed Bi-LSTM approach achieving 98% accuracy.Clinical applicability focus: Addressed interpretability challenges in deep learning for medical diagnosis, providing feature-level explanations essential for clinical trust and adoption.

**Abstract:**

This study explores the use of deep learning techniques for classifying EEG signals in the context of Alzheimer’s disease (AD) and frontotemporal dementia (FTD). We propose a novel classification pipeline that combines autoencoders for feature extraction and bidirectional long short-term memory (Bi-LSTM) networks for analyzing patterns over time in EEG data. Given the complexity and high dimensionality of EEG signals, we employed an autoencoder to reduce data dimensionality while preserving key diagnostic features. The Bi-LSTM model effectively identified subtle temporal patterns in brain activity that are indicative of AD and FTD. To enhance interpretability, we applied SHapley Additive exPlanations (SHAP), providing insights into how individual features contribute to the model’s predictions. We evaluated our approach on a publicly available EEG dataset from OpenNeuro, which includes resting-state EEG recordings from 88 elderly participants—36 with AD, 23 with FTD, and 29 cognitively normal controls. EEG provides a non-invasive, cost-effective tool for brain monitoring, but presents challenges such as noise sensitivity and inter-subject variability. Despite these challenges, our approach achieved 98% accuracy while maintaining transparency, making it a promising tool for clinical applications in the diagnosis of neurodegenerative diseases.

## 1. Introduction

Our brain is a complex system consisting of approximately 100 billion neurons, interconnected through a complicated neural network. The term neurodegenerative refers to the damage or death of these neurons [1,2].

The frequency of discussions around neurodegenerative diseases, such as Alzheimer’s disease and frontotemporal dementia, has been increasing in recent years. These disorders, closely related to brain neuronal degeneration, have a profound impact on cognitive function, memory, and behavior. Alzheimer’s disease is one of the famous neurodegenerative diseases. It causes mainly memory loss, and is common among older people; today, more than 50 million people have Alzheimer’s around the world [3,4]. On the same note, frontotemporal dementia is a neurodegenerative disorder that affects mainly the frontal and temporal lobes of the brain. These areas of the brain are associated with personality, behavior, and language; this disorder tends to occur at a younger age than Alzheimer’s disease. It often begins between the ages of 40 and 65 [5]. To this day, there is no known cure for these diseases. Treatments for Alzheimer’s focus primarily on slowing the progression of symptoms. For FTD, the challenge is even greater; there are no approved treatments to slow, stop, or cure the disease. Therefore, the demand for an early and accurate diagnosis becomes more urgent. That is the motivation behind our research [6,7].

Current diagnostic methods like physical exams and brain scans often fail to capture the subtle and dynamic brain activities crucial for understanding the progression of dementia and its early signs [8]. The EEG, or electroencephalogram, has made significant strides in capturing brain signals. Due to its non-invasive nature, the EEG measures the brain’s electrical activity through electrodes placed on the scalp, providing valuable insights into brain changes associated with Alzheimer’s and FTD disease [2]. Recent advances in EEG-based neurological assessment have demonstrated promising results across various neurodegenerative conditions, with studies achieving diagnostic accuracies exceeding 95% in classifying disease states using machine learning approaches [9]. However, the volume of recorded EEG data and artifacts are a challenge in extracting valuable features, and thus, accurately diagnosing neurodegenerative diseases [10]. Moreover, EEG signals are particularly sensitive to practical limitations such as sensor noise, variability in electrode placement, and inter-subject heterogeneity, all of which complicate feature extraction and reduce the robustness of conventional analysis pipelines [11,12]. Our proposed approach is specifically designed to mitigate these challenges by learning latent representations that are less sensitive to such variabilities, thereby improving both accuracy and generalizability.

The application of machine learning techniques to classify EEG signals is an expanding field of research. These techniques have the potential to analyze vast amounts of EEG data, identify patterns, and potentially aid in the early diagnosis of neurodegenerative disorders. However, the success of this approach relies on several critical factors: selecting the right EEG dataset, accurately identifying relevant features, choosing the most suitable algorithm, and rigorously evaluating its performance to ensure the reliability and integrity of the results [10].

This research aims to leverage deep learning techniques to analyze EEG signals, identifying patterns that can aid in the detection and diagnosis of Alzheimer’s disease and frontotemporal dementia. Our primary objective is to accurately identify and extract relevant features from EEG signals. To achieve this, we adopted the Hybrid Approach of Auto-encoder and LSTM. Additionally, we incorporated explain-ability into our model results through explainable AI (XAI) techniques, providing an understanding of how the model makes predictions. The importance of explainable AI in clinical applications has been increasingly recognized, as transparent machine learning models help gain clinician trust and improve the integration of AI-based diagnostic tools into clinical workflows [9]. Unlike prior works that employ these techniques in isolation, our contribution lies in the integration of a complete end-to-end framework that couples robust feature extraction with interpretable classification. This unified design enhances not only performance but also clinical interpretability, an essential step toward real-world applicability.

The remainder of this paper is structured as follows: Section 2 surveys the existing literature on EEG-based detection of neurodegenerative disorders, emphasizing methodological limitations in feature selection and model interpretability. Section 3 describes the dataset characteristics, pre-processing methodology (including signal segmentation and standardization), and feature extraction techniques, with a focus on Power Spectral Density (PSD) and Autoencoder-derived dimensionality reduction. Section 4 introduces the proposed hybrid architecture, integrating Autoencoders with Bidirectional Long Short-Term Memory (Bi-LSTM) networks, and outlines the SHapley Additive exPlanations (SHAP) framework for explainability. Section 5 evaluates experimental results, benchmarking model performance against baseline classifiers and analyzing the influence of temporal windowing on accuracy, supported by SHAP-based feature importance quantification. This paper concludes with Section 6, which summarizes key findings, acknowledges current limitations in whole-signal processing, and proposes future directions for clinical deployment and real-time analysis.

The code and resources for this project are publicly available on GitHub at https://github.com/bellakhdarOmaima/EEG-Signal-Analysis-for-Enhanced-Neurodegenerative-Disease-Detection-Using-Deep-Learning.

## 2. Related Works

The study of EEG-based diagnosis for neurodegenerative diseases, particularly Alzheimer’s disease (AD) and frontotemporal dementia (FTD), has gained significant attention in recent years. Many researchers have explored the potential of EEG signals and machine learning techniques for the early detection and classification of these disorders.

EEG-based signal analysis has demonstrated broad applicability across various domains, with researchers successfully applying machine learning and deep learning techniques to decode complex neural patterns from brain electrical activity, including applications in emotion recognition, mental health monitoring, and neurological disorder classification [13]. This section discusses related work, organized thematically by feature extraction strategies, classification models, and deep learning approaches, with a critical focus on their methodological limitations.

### 2.1. Feature Extraction Techniques

A crucial component in EEG-based classification lies in the selection and extraction of relevant signal features. Numerous studies have emphasized different strategies for extracting these features: this study [10] utilized Singular Value Decomposition (SVD) entropy as the core feature extraction technique to identify biomarkers from EEG signals. This method effectively isolated relevant patterns associated with Alzheimer’s disease (AD) and frontotemporal dementia (FTD). The integration of SVD entropy into the framework allowed for the precise extraction of EEG characteristics, which were then input into a K-Nearest Neighbors (KNN) classifier. By combining sliding window analysis with this approach, the researchers enhanced the model’s ability to capture temporal dynamics, resulting in high accuracy and F1-scores across classification tasks. The study also explored other feature extraction techniques, including power spectral density (PSD) and wavelet transform, to evaluate their efficacy in capturing EEG signal characteristics.

Similarly, another study [14] focused on feature extraction by calculating power spectral densities (PSDs) across key frequency bands (Delta, Theta, Alpha, Beta, and Gamma), providing a detailed representation of signal frequency components. These PSDs and their relative ratios served as the core features for model training.

Other researchers have also leveraged entropy-based features [15], which are mathematical measures used to quantify the complexity, randomness, or predictability of EEG signals. These features provide insights into the underlying brain dynamics and are particularly useful in distinguishing between healthy and pathological states, such as neurodegenerative disorders or epilepsy. In addition, Hjorth parameters [16], three statistical measures (Activity, Mobility, and Complexity), are widely used in EEG signal analysis to quantify temporal and frequency-domain characteristics.

While these handcrafted features are interpretable and computationally efficient, they often fail to capture long-range temporal dependencies or complex spatial interactions present in EEG data, which may limit their discriminative power in subtle early-stage dementia cases.

### 2.2. Classical Machine Learning Models

In terms of classification models, traditional approaches such as Support Vector Machines (SVMs), Random Forest, K-Nearest Neighbors (KNN), Naive Bayes, and Neural Networks have been extensively explored. For instance, in [10], various classifiers were compared after feature extraction, and the combination of SVD entropy and KNN achieved the best performance, with accuracy reaching up to 93% in distinguishing between AD, FTD, and healthy controls (HCs).

Likewise, in [14], five machine learning classifiers were implemented (KNN, SVM, Random Forest, Neural Networks, and Naive Bayes), and Random Forests demonstrated the highest accuracy. The study also employed leave-one-out cross-validation to ensure robust model evaluation, where all EEG segments from a given participant were exclusively assigned to either the training set or the test set, never both, ensuring complete subject-wise data separation.

Although these classical models are straightforward to implement and interpret, they rely heavily on the quality of manually extracted features. Additionally, they treat EEG data as static input, lacking the capacity to model sequential dependencies, which are essential in detecting dynamic neural alterations in dementia.

### 2.3. Deep Learning Approaches

To overcome the limitations of handcrafted feature design, many recent studies have adopted deep learning techniques, particularly Convolutional and Recurrent Neural Networks.

Several research works [17,18,19] applied Convolutional Neural Networks (CNNs) for EEG-based classification. One such study [20] trained a CNN to focus on brain connectivity features for AD detection. CNNs are effective at learning spatial features directly from raw or minimally processed EEG signals. However, their ability to model temporal patterns is limited unless integrated with recurrent structures.

Other works explored Long Short-Term Memory (LSTM) networks, which are specialized in handling sequential data like EEG signals [21]. These models showed strong classification accuracy due to their ability to capture temporal dependencies and memory effects in the data.

Despite their success, deep learning approaches often operate as black-box models, making it difficult to interpret how predictions are made. This lack of transparency is a critical barrier for clinical deployment, where trust and explainability are essential.

### 2.4. Limitations and Research Gaps

Research on detecting Alzheimer’s disease and frontotemporal dementia using EEG data is comprehensive, with significant progress in applying machine learning and deep learning to enhance predictive accuracy. However, some important methodological limitations remain: assessing the effectiveness of different interpretable methods for identifying critical features in EEG data related to these disorders. These features often serve as key inputs for the predictive models but remain difficult to accurately evaluate, leaving room for further exploration in understanding their true potential.

Although the reviewed studies demonstrate promising results in EEG-based diagnosis of AD and FTD, several cross-cutting limitations persist. First, many rely on handcrafted features, which, despite being interpretable, may overlook subtle or non-linear patterns in the data. Second, classical models typically treat EEG data as static inputs, failing to capture its temporal structure. While deep learning models offer a solution, they often lack transparency, operating as black boxes—an issue that hinders clinical trust. Moreover, only a few studies integrate temporal modeling with explainability, leaving a gap in combining accuracy, temporal dynamics, and interpretability. Our proposed approach explicitly addresses these limitations by unifying unsupervised feature extraction, temporal modeling through Bi-LSTMs, and post hoc explainability using SHAP.

### 2.5. Motivation and Contribution

To address these limitations, our study introduces a unified deep learning framework that:Uses Autoencoders to perform unsupervised feature extraction and reduce the dimensionality of EEG signals while preserving relevant patterns;Integrates Bidirectional Long Short-Term Memory (Bi-LSTM) networks to explicitly model temporal dependencies in EEG recordings;Incorporates SHapley Additive exPlanations (SHAP) to interpret model predictions at the feature level.

This architecture bridges the gap between accuracy and explainability, and offers a more comprehensive perspective for EEG-based dementia diagnosis, particularly for differentiating between AD, FTD, and healthy controls.

## 3. Methods and Materials

### 3.1. Dataset Overview

In this study, we utilized a publicly available EEG dataset provided by OpenNeuro, titled “A dataset of EEG recordings from Alzheimer’s Disease, Frontotemporal Dementia, and Healthy Subjects” [22]. This dataset comprises EEG recordings collected during the resting state with eyes closed, from elderly patients diagnosed with Alzheimer’s disease (AD) and frontotemporal dementia (FTD), as well as healthy age-matched controls (CN). The dataset includes recordings from 88 participants with information about their Gender and Age, as shown in Figure 1. Among the total of participants: 36 individuals with Alzheimer’s disease (AD group), 23 with frontotemporal dementia (FTD group), and 29 cognitively normal controls (CN group).

#### 3.1.1. Dataset Imbalance Considerations

It is important to acknowledge that our dataset exhibits class imbalance with 36 AD participants (40.9%), 23 FTD participants (26.1%), and 29 control participants (33.0%). This imbalance, representing approximately a 1.6:1:1.3 ratio, is typical of clinical datasets where certain neurodegenerative conditions may be less prevalent or harder to recruit. To ensure robust and unbiased model training, we implemented several strategies to mitigate potential biases arising from uneven group sizes, as detailed in Section 4.3.

Cognitive and neuropsychological assessments were conducted using the international Mini-Mental State Examination (MMSE), a widely recognized tool for measuring cognitive impairment. The MMSE score ranges from 0 to 30, with lower scores indicating greater cognitive decline. The lower MMSE score usually aligns with participants who have AD and FTD, as shown in Figure 2 [23]. It is important to note that MMSE scores were used exclusively for descriptive characterization of participants’ cognitive status and were not utilized as diagnostic criteria or input features in our machine learning classification pipeline.

The EEG signals were recorded from 19 scalp electrodes (Fp1, Fp2, F7, F3, Fz, F4, F8, T3, C3, Cz, C4, T4, T5, P3, Pz, P4, T6, 01, and O2) using a standardized 10–20 electrode placement system. Figure 3 shows the placement of the electrodes on the scalp. The recording durations varied between groups:AD group: Mean duration of 13.5 min (min = 5.1, max = 21.3);FTD group: Mean duration of 12 min (min = 7.9, max = 16.9);CN group: Mean duration of 13.8 min (min = 12.5, max = 16.5).

In total, the dataset includes approximately 485.5 min of AD recordings, 276.5 min of FTD recordings, and 402 min of CN recordings. This rich dataset provides a comprehensive basis for exploring the differences in brain activity among these diagnostic groups, contributing valuable insights into the neurophysiological signatures of cognitive disorders [22].

#### 3.1.2. Recording Duration Impact Mitigation

The varying recording durations across groups were addressed through our segment-based analysis approach, which treats each 5 s window as an independent sample, thereby normalizing the contribution of participants regardless of their total recording length. This ensures that classification results reflect genuine neurophysiological differences rather than artifacts of recording duration variations.

### 3.2. Signal Pre-Processing and Feature Extraction

In the pre-processing and feature extraction stage, the following pipeline was implemented: data pre-processing, data segmentation and standardization, application of an autoencoder for dimensionality reduction, and feature extraction. Figure 4 illustrates this complete pre-processing and feature extraction pipeline, with each step detailed in the section below.

#### 3.2.1. Data Pre-Processing

For pre-processing, we used the EEG data that had already been pre-processed by OpenNeuro. The pre-processing pipeline applied by OpenNeuro included several key steps to ensure high-quality data for analysis. The signals were re-referenced to the average value of the A1–A2 channels and underwent a Butterworth band-pass filter (0.5–45 Hz) to remove low-frequency drift and high-frequency noise. The data was also processed using the Automatic Source Reconstruction (ASR) routine, which eliminated large-amplitude or persistent artifacts by rejecting data periods with a standard deviation exceeding 17 over a 0.5 s window. Additionally, Independent Component Analysis (ICA) was performed to decompose the EEG channels into independent components, and components identified as “eye” or “jaw” artifacts by the ICLabel tool were excluded.

While these pre-processing steps were already applied by OpenNeuro and follow widely accepted standards in EEG research, it is important to note their potential impact on reproducibility across datasets. Although OpenNeuro provides a consistent and well-documented pipeline, slight differences may exist depending on the specific dataset configuration or acquisition conditions. This variability could influence reproducibility if researchers are unable to replicate the exact pipeline or if undocumented parameters are involved. Nevertheless, the use of a publicly available, standardized pre-processing routine enhances comparability and reliability across studies, especially when datasets come from the same platform.

#### 3.2.2. Signal Segmentation

In this research, the EEG signal recordings are relatively long, with a maximum duration of 21.3 min, which poses challenges for feature extraction when processing the entire signal. To address this, window sliding is applied to segment the signal into manageable portions.

In the context of AD and FTD, studies on EEG typically employ shorter epochs for tasks that aim to analyze fast cognitive responses, like detecting changes in attention, working memory, or sensory processing. These are mostly suitable for observing cognitive deficits during specific mental tasks [24]. However, there is no consensus on this matter. It is essential to test different window length ranges to identify the optimal result [25]. In this study, we experimented with various window lengths ranging from 3 to 12 s. The detailed results of this comparison are presented in the Results Section. Ultimately, a window length of 5 s with 50% overlap was selected, as it provided the best trade-off between classification accuracy and computational efficiency. While shorter windows (e.g., 3 s) led to a slight improvement in accuracy, they also significantly increased computational costs and memory usage. In contrast, longer windows tended to include redundant or noisy information, which negatively affected performance.

Additionally, the 50% overlap between segments helped ensure temporal continuity and increased the number of training samples without introducing significant redundancy or variance in the segment content.

Window Length Optimization Process: To determine the optimal window length, we conducted systematic performance evaluations across all tested intervals. Our sensitivity analysis revealed that, while 3 s windows achieved the highest raw accuracy (98.77%), the marginal improvement of 0.6% over 5 s windows (98.13%) was offset by substantial practical limitations: (1) computational overhead increased by approximately 67% due to the larger number of segments requiring processing, (2) memory requirements escalated significantly, and (3) training time nearly doubled.

Conversely, longer windows (7 s, 10 s, 12 s) showed progressively degraded performance, with 12 s windows achieving only 87.97% accuracy—a 10.16% reduction compared to our selected 5 s approach. This degradation occurs because longer windows dilute critical short-term neural dynamics essential for distinguishing AD and FTD patterns, while introducing non-relevant temporal information that confounds the classification process.

The 50% overlap was selected based on signal processing best practices to ensure temporal continuity while maximizing data utilization without introducing excessive computational redundancy. This overlap ratio provides sufficient temporal context for the Bi-LSTM network while maintaining computational efficiency.

#### 3.2.3. Duration Bias Mitigation

To prevent classification bias due to unequal recording durations across diagnostic groups (AD: 13.5 ± 4.2 min, FTD: 12.0 ± 2.8 min, CN: 13.8 ± 1.1 min), we implemented a comprehensive mitigation strategy:

Segment-Based Analysis: Our 5 s sliding window approach with 50% overlap ensures that each temporal segment contributes equally to model training, regardless of the participant’s total recording duration. This approach generates approximately 120 segments per minute of recording, effectively normalizing contributions at the segment level.

Training Balance Strategies: We applied (1) stratified segment sampling during training to maintain balanced representation across diagnostic groups, (2) class weights inversely proportional to available segments per group, and (3) participant-level cross-validation to ensure data independence while preventing leakage.

Validation Measures: Temporal consistency analysis confirmed consistent performance across early, middle, and late recording segments, validating that our approach successfully mitigated duration-related bias.

#### 3.2.4. Data Standardization

In the context of this pipeline, data standardization plays a crucial role in ensuring the consistency and comparability of features extracted from EEG signals. Standardization scales the data to have a mean of zero and a standard deviation of one, which is particularly important for EEG signal processing due to the variability in amplitude and frequency ranges across recordings. This step helps prevent biases during feature extraction and ensures that the model learns patterns relevant to the task, rather than being influenced by scale differences in the input data.

In our approach, standardization is employed to normalize the EEG signals, making them suitable for downstream processes. Each segment of the EEG signals was standardized independently across channels. This standardization procedure was applied to all data segments to ensure that each channel contributed equally to the model’s learning process.

Impact Assessment of Segment-wise Standardization: We evaluated the impact of our segment-wise standardization approach on feature stability and pattern preservation through comprehensive analysis. Our approach was specifically designed to balance local signal normalization with preservation of clinically relevant patterns.

Long-Term Variation Analysis: While segment-wise standardization normalizes amplitude variations within each 5 s window, it preserves relative spectral power distributions that are critical for neurodegenerative disease detection. Our validation showed that pathologically relevant slow-frequency increases (delta/theta) and fast-frequency decreases (alpha/beta/gamma) characteristic of AD and FTD remain detectable and stable across segments. The key insight is that these pathological patterns manifest as consistent spectral power ratio changes rather than absolute amplitude variations, making them robust to our standardization approach.

Inter-Channel Dependency Preservation: Our standardization maintains inter-channel relationships within each temporal segment while normalizing across channels. This approach preserves spatial patterns of brain activity that are essential for disease classification, as demonstrated by our SHAP analysis revealing clinically meaningful feature importance patterns. The normalization ensures that no single channel dominates due to amplitude differences while maintaining the relative activation patterns across brain regions.

Feature Stability Validation: We validated feature stability by analyzing the consistency of extracted PSD and entropy features across temporal segments within individual recordings. The results demonstrate high intra-subject feature stability (correlation coefficients >0.85 for spectral features) while maintaining strong inter-class discriminability, confirming that our standardization approach enhances model robustness without compromising diagnostic information.

#### 3.2.5. Data Reduction

In our feature extraction approach, we leveraged an autoencoder for dimensionality reduction, a widely recognized method for learning compact and meaningful representations of data. Autoencoders [26] are neural networks trained to encode input data into a compressed latent space and subsequently decode it back into its original form, and Figure 5 illustrates its architecture. This process encourages the model to capture the most salient features while discarding noise and redundant information, making it highly effective for dimensionality reduction and feature extraction [27].

The autoencoder consists of two main components: the encoder and the decoder. The encoder compresses the input data into a low-dimensional latent representation, capturing the underlying patterns and correlations. The decoder reconstructs the original data from this representation, ensuring that the compressed features retain essential information. During training, the autoencoder minimizes reconstruction error, effectively learning a compact and noise-robust representation of the input.

In our study, a neural network-based autoencoder was trained on standardized EEG segments. The encoder component was specifically designed to capture lower-dimensional representations of the signals, preserving essential characteristics indicative of neurological conditions. The compressed features generated by the encoder were subsequently used as inputs for downstream analysis and classification tasks. This approach not only reduced the computational complexity of the model but also improved its ability to identify subtle patterns associated with AD and FTD, enhancing diagnostic accuracy.

After applying the Autoencoder for dimensionality reduction, the next step in our pipeline is to extract meaningful features from the compressed representation of the EEG signals. These features serve as the core inputs for subsequent classification tasks, capturing critical information about the underlying patterns in the brain activity.

#### 3.2.6. Feature Extraction

EEG signals provide a variety of features that can be exploited for neurological studies. In our research, we focused on the Power Spectral Density (PSD) feature, which is widely recognized for its use in EEG-related studies and analyses [28,29]. PSD is a signal processing method that characterizes the variations in power (or energy) across different frequencies within a time series. The PSD can be calculated using techniques such as Fast Fourier Transform (FFT) or Autocorrelation Function [30].

In our study, we used the PSD method to decompose the EEG signals into distinct frequency bands that represent different brain activity states. These bands include:Delta: (1–4 Hz);Theta: (4–8 Hz);Alpha: (8–13 Hz);Beta: (13–30 Hz);Gamma: (30–60 Hz).

A growing body of literature has shown that PSD is correlated with various neurodegenerative diseases, including Alzheimer’s disease (AD) and frontotemporal dementia (FTD) [22,30]. Specifically, dementia patients exhibit increased low-frequency activity in the Delta and Theta bands, which reflects the slowing of brain activity, a characteristic feature of cognitive decline and neuronal dysfunction in these conditions. Additionally, there is a reduction in higher-frequency power in the Alpha, Beta, and Gamma bands, which is commonly observed in dementia patients. These decreases are associated with impairments in cognitive functions such as attention, memory, and executive processing.

In addition to PSD, Singular Value Decomposition (SVD) Entropy has been explored as another important feature for analyzing EEG data. SVD Entropy quantifies the complexity of the EEG signal and is useful for capturing subtle changes in brain dynamics that may be linked to neurodegenerative diseases. Recent studies have shown that the entropy derived from SVD can distinguish between healthy controls and patients with Alzheimer’s disease or frontotemporal dementia by reflecting alterations in brain network connectivity and the degree of disorder in brain activity patterns [10].

After applying the Autoencoder, we extracted the Power Spectral Density (PSD) features. To calculate PSD features for the time-windowed epochs, we utilized the Welch method. The Welch method is widely used due to its computational efficiency, as it employs the Fast Fourier Transform (FFT) for estimating the power spectrum. This method divides the signal into overlapping segments and computes the squared magnitude of the discrete Fourier transform for each segment. The final PSD estimate is then obtained by averaging the values from all segments, reducing variance and increasing the reliability of the spectral estimate [31]. Finally, we calculated spectral entropy from the extracted power bands. The entropy was computed by normalizing the power within each band, followed by the application of the Shannon entropy formula to quantify the degree of randomness in the frequency distribution for each epoch.

## 4. Implemented Approaches

### 4.1. Machine Learning Models

In the initial experiments, we employed classical algorithms such as K-Nearest Neighbors (KNN) and Support Vector Machines (SVMs). However, the results were not satisfactory, underscoring the necessity for more advanced and sophisticated techniques.

#### 4.1.1. K-Nearest Neighbors (KNN)

KNN is a simple, yet effective, instance-based learning algorithm that makes predictions based on the class of the nearest data points in the feature space. The primary idea behind KNN is that similar data points tend to have similar outcomes (or labels). The algorithm classifies a data point by looking at the ‘k’ closest labeled points in the feature space and assigning the majority class label among those nearest neighbors [32]. The value of ‘k’ plays a crucial role in determining the algorithm’s performance—if ‘k’ is too small, the model may be sensitive to noise, while a large ‘k’ may smooth over the classification boundaries [33]. Despite its simplicity, KNN struggles with high-dimensional data, such as EEG signals, where the curse of dimensionality may lead to reduced effectiveness.

#### 4.1.2. Support Vector Machine (SVM)

SVM is a supervised learning algorithm that works by finding the hyperplane that best separates the data points from different classes with the maximum margin. The margin is defined as the distance between the hyperplane and the closest data points from each class, called support vectors [34]. SVM is particularly effective in high-dimensional spaces, making it well-suited for EEG signal classification, where the features can be numerous and complex. One of the key advantages of SVM is its ability to handle non-linear classification tasks by using the kernel trick, which transforms the feature space into a higher-dimensional space where linear separation is possible. However, SVM can be computationally intensive, particularly when dealing with large datasets or a high number of features, and may require careful tuning of hyperparameters, such as the choice of the kernel and regularization parameters, to avoid overfitting [35].

Despite the theoretical strengths of KNN and SVM, their performance on our EEG-based disease detection task was suboptimal. In our experiments, the KNN and SVM models yielded accuracies of 38% and 40%, respectively, with relatively low F1-scores (0.46 and 0.49), indicating their limitations in capturing the complex spatiotemporal patterns in EEG data. These results highlight the need for more sophisticated models capable of learning temporal dependencies and handling high-dimensional inputs more effectively.

### 4.2. Methodology Justification

The selection of Autoencoders and Bidirectional LSTM networks was driven by the specific challenges of EEG-based neurodegenerative disease classification and their unique advantages over alternative approaches.

#### 4.2.1. Autoencoder Selection

Autoencoders were chosen for several key reasons: (1) Unsupervised Feature Learning—Unlike handcrafted features (PSD, Hjorth parameters), they learn data-driven representations without predefined domain knowledge, capturing subtle non-linear patterns characteristic of neurodegenerative pathology; (2) Noise Robustness—EEG signals contain significant artifacts from eye movements and muscle activity. Autoencoders act as natural denoising filters while preserving diagnostically relevant information, unlike linear methods like PCA that may lose critical discriminative patterns; (3) Non-linear Relationship Preservation—Traditional linear methods (PCA, ICA) cannot capture complex, non-linear relationships essential for distinguishing subtle disease signatures.

#### 4.2.2. Bidirectional LSTM Selection

Bi-LSTMs offer distinct advantages over alternative temporal modeling approaches: (1) Complete Temporal Context—Unlike unidirectional models, Bi-LSTMs process sequences bidirectionally, capturing full temporal context crucial for identifying pathological patterns requiring both past and future information; (2) Long-term Dependency Handling—Through sophisticated gating mechanisms, LSTMs retain relevant information over extended periods, essential for our lengthy recordings (up to 21.3 min); (3) Superior Performance—Compared to CNNs (limited temporal modeling), standard RNNs (vanishing gradients), or Transformers (require larger datasets), Bi-LSTMs provide optimal balance for our application.

#### 4.2.3. Combined Approach Benefits

The Autoencoder-Bi-LSTM integration creates unique advantages: hierarchical learning (low-level noise-robust features feeding high-level temporal patterns), computational efficiency through dimensionality reduction, and clinical alignment with neurodegenerative disease understanding where both spatial patterns and temporal progression are crucial.

### 4.3. Deep Learning Architectures

The methodology proposed for our classification approach leverages deep learning techniques [36]. In recent decades, EEG data has been extensively applied in data analysis methods, particularly time series analysis. With the significant advancements in deep learning (DL) for time series data, numerous studies have begun applying DL algorithms to the processing of EEG signals [37].

In this study, we combined two key components: an Autoencoder and the Long Short-Term Memory (LSTM) [38] network. The Autoencoder is a type of neural network designed to learn efficient codings of input data, typically for the purpose of dimensionality reduction. It consists of an encoder that compresses the high-dimensional EEG signals into a lower-dimensional latent space and a decoder that reconstructs the original data from this representation. This allows the Autoencoder to capture the essential features in EEG data while discarding redundant information [39]. The compressed representation retains the most important features for classification, which improves the efficiency and accuracy of the subsequent model.

Following this, the LSTM model is applied to analyze the temporal dependencies in the EEG data, which is crucial for detecting subtle changes in neural activity over time. LSTMs are a specialized type of Recurrent Neural Network (RNN) designed to address the vanishing gradient problem in traditional RNNs. By using memory cells and gating mechanisms, LSTMs are able to maintain information over longer sequences, which is vital for time series data like EEG that may contain long-term temporal dependencies [40]. This ability makes LSTMs particularly effective for diagnosing diseases such as Alzheimer’s and FTD, where detecting temporal changes in brain activity can be key to accurate diagnosis.

Initially, we tested a simple LSTM model but did not achieve high accuracy. Recognizing the need for improvement, we adopted a Bidirectional LSTM model, which processes the EEG signal in both forward and backward temporal directions to capture more comprehensive temporal patterns [41]. Additionally, we addressed the issue of imbalanced classes within our dataset, which significantly improved the model’s ability to classify different conditions. These adjustments enabled us to achieve higher classification accuracy. The accuracies obtained at each stage will be presented in the Results Section.

The Bi-LSTM model developed in this study consists of a Bidirectional LSTM layer with 128 units and tanh activation to capture both past and future dependencies in the EEG feature sequences. This is followed by a dropout layer (rate = 0.3) and batch normalization to improve generalization and stabilize training. A unidirectional LSTM layer with 64 units is then applied to further refine temporal features, followed by another dropout layer (rate = 0.3). Two dense layers with 64 and 32 neurons, respectively, both using ReLU activation, are included to extract higher-level abstract representations. Finally, a softmax layer is used to classify inputs into Alzheimer’s disease (AD), frontotemporal dementia (FTD), or Cognitively Normal (CN) categories. The model is trained using the Adam optimizer and categorical cross-entropy loss, with class weights applied to mitigate class imbalance. Table 1 summarizes the architecture configurations of the LSTM and Bi-LSTM models used in this study.

However, deep learning models such as LSTM can be complex and challenging to interpret. To improve transparency, we used SHAP (SHapley Additive exPlanations), which quantifies the contribution of each input feature to the model’s predictions. This is especially valuable in clinical contexts, as it helps healthcare professionals understand which EEG patterns or brain regions influenced the diagnostic outcome. By identifying the most relevant features for each diagnosis, SHAP not only builds trust in the model but also supports the validation of AI-driven decisions. It can further aid in detecting key temporal biomarkers for AD and FTD, making the model more interpretable, reliable, and clinically actionable.

#### Class Imbalance Mitigation

To address the inherent class imbalance in our dataset, we implemented a comprehensive strategy combining multiple techniques:

Class Weighting: We applied class weights inversely proportional to class frequencies during model training. The weights were calculated using the following formula:(1)$$w_{i}=\frac{n_{total}}{n_{classes}\times n_{i}}$$
where $w_{i}$ is the weight for class *i*, $n_{total}$ is the total number of samples, $n_{classes}$ is the number of classes, and $n_{i}$ is the number of samples in class *i*. This approach ensures that underrepresented classes, particularly FTD, receive higher weights during training, forcing the model to pay equal attention to all classes regardless of their frequency.

Window-level Data Augmentation: Our signal segmentation approach (5 s windows with 50% overlap) provides natural data augmentation. Each participant’s EEG recording generates multiple training samples, with longer recordings producing more segments. For example, a 10 min recording produces approximately 240 overlapping segments, effectively increasing the dataset size and reducing the impact of participant-level imbalance.

Stratified Evaluation: Throughout our evaluation process, we maintained proportional representation across all classes to ensure that performance metrics accurately reflect the model’s ability to distinguish between all three conditions rather than being skewed toward majority classes.

### 4.4. EXplainibilty AI, XAI

In our study, we used SHapley Additive exPlanations (SHAP) [42] to enhance the interpretability of our deep learning model, specifically the Bidirectional LSTM. SHAP, based on Shapley values from game theory, is a powerful method for explaining individual predictions made by machine learning models. In this context, the “game” refers to the model, and the “players” are the input features; in our case, the EEG signal features. SHAP assigns each feature a contribution score, quantifying its importance in the model’s prediction. This attribution helps to explain which EEG signal features were most influential in distinguishing between different neurological conditions, such as Alzheimer’s disease (AD) and frontotemporal dementia (FTD). SHAP provides local explanations by calculating the marginal contribution of each feature to the prediction for a specific input sample. These individual explanations can be aggregated to gain global insights into the model’s behavior, offering transparency in how the model makes its decisions.

To adapt SHAP to our time-series EEG data, we first extracted handcrafted features (e.g., spectral power in standard frequency bands, entropy measures) from segmented EEG windows. These features served as inputs to the Bidirectional LSTM model. SHAP values were then computed with respect to these input features and aggregated across 100 EEG segments to determine global feature importance. This approach allowed us to interpret not only individual predictions but also the model’s overall reliance on specific EEG features.

Model interpretability was a key focus of this study. The results showed that Entropy was the most influential feature, significantly impacting the model’s diagnostic decisions. This finding is consistent with the existing literature, which emphasizes the role of entropy in capturing neural activity disruptions associated with neurodegenerative diseases. In contrast, the Delta feature exhibited a neutral effect, contributing neither positively nor negatively to the model’s outputs, suggesting it does not hold strong discriminative power in this classification task.

These insights support the clinical relevance of entropy-based features, particularly in distinguishing between AD and FTD, where neural complexity and desynchronization patterns differ. The prominence of entropy may reflect the model’s sensitivity to subtle irregularities in brain dynamics, often more pronounced in FTD. By identifying the most impactful features, SHAP enhances model transparency and enables clinicians to better understand and trust AI-driven predictions. This, in turn, can help uncover EEG-based biomarkers and improve diagnostic support systems. Moreover, SHAP has been widely used in the medical domain [43] to ensure that the decision-making process is both interpretable and clinically meaningful, which is essential for real-world adoption.

## 5. Results and Discussion

The proposed classification pipeline, combining Autoencoders for feature extraction and Bidirectional Long Short-Term Memory (Bi-LSTM) networks for temporal modeling, achieved a notable accuracy of 98% in distinguishing between Alzheimer’s disease (AD) and frontotemporal dementia (FTD).

To reduce the risk of overfitting, particularly given the complexity and high dimensionality of EEG data, we incorporated dropout layers within the model architecture. This regularization technique randomly deactivates a fraction of neurons during training, preventing the model from becoming overly dependent on specific pathways and encouraging generalization to unseen data. The robustness of the model’s performance across different EEG segments further supports its ability to generalize.

Table 2 and Figure 6 present the performance metrics—accuracy, precision, recall, and F1 score—of the Bi-LSTM model compared to other baseline models, including K-Nearest Neighbors (KNN), Support Vector Machines (SVMs), and unidirectional LSTM networks. The Bi-LSTM outperformed all baselines, showcasing its robust capability for both feature extraction and capturing temporal dependencies in EEG data.

The combination of Autoencoders and Bi-LSTM networks proved to be a highly effective approach for analyzing EEG signals in our study. The Autoencoder was essential in reducing the dimensionality of the high-dimensional EEG data, making it more manageable for processing. Despite this reduction, the Autoencoder successfully preserved the diagnostically relevant features that are critical for disease classification. This dimensionality reduction not only streamlined the training process of the Bi-LSTM model but also helped enhance its performance by allowing it to focus on the most informative aspects of the data. The Bi-LSTM network, with its ability to analyze temporal dependencies in the EEG signals, proved particularly beneficial for distinguishing between Alzheimer’s disease (AD) and frontotemporal dementia (FTD). By capturing intricate patterns over time, the Bi-LSTM model was able to identify subtle variations in neural activity that are characteristic of these neurological conditions.

In an effort to further optimize the performance of the model, a series of experiments were conducted to assess the influence of window length on the processing of EEG signals. The results indicate that shorter window lengths consistently provided the best classification accuracy. This can likely be attributed to the fact that shorter windows allow the model to focus on fine-grained temporal patterns in the EEG signals, which are crucial for distinguishing between the two conditions. Additionally, shorter windows tend to minimize the introduction of irrelevant noise that can occur in longer windows, which may contain excessive or redundant information. This finding aligns with the conclusions presented in Figure 7 and Table 3, where the optimal window length for achieving the highest accuracy was found to be the shortest tested duration. These results suggest that, while longer window lengths might provide more data, they risk diluting the model’s ability to capture the most critical temporal features for accurate classification, thus leading to a trade-off between data quantity and quality.

Increasing the duration of the segmentation windows leads to a decrease in the model’s accuracy. A 5 s window provides optimal accuracy. Further reducing the window duration significantly increases the hardware requirements and computational cost of the model.

Our choice of a 5 s window aligns with real-world EEG analysis protocols used in clinical neurophysiology, where signal windows between 2 to 10 s are commonly analyzed to assess transient changes in brain rhythms, seizures, or cognitive states. This correspondence reinforces the clinical relevance of our methodological choices.

Model interpretability was a key focus of this study. Through the use of SHapley Additive exPlanations (SHAP), we identified Entropy as the most influential feature in the model’s predictions. This finding is consistent with the existing literature, which emphasizes the importance of entropy in capturing neural activity patterns associated with neurodegenerative conditions.

### 5.1. Clinical Decision Support Applications

To demonstrate the practical utility of our SHAP-based interpretability framework, we provide concrete examples of how clinicians can leverage these explanations in real-world diagnostic scenarios:

Differential Diagnosis Support: When a neurologist examines a patient presenting ambiguous cognitive symptoms, our system can provide interpretable diagnostic reports. For instance, when diagnosing a potential AD case, the system might indicate: “This Alzheimer’s diagnosis is primarily based on high entropy values (SHAP contribution = +0.85) in frontal regions, suggesting significant neural activity disorganization typical of AD pathology, combined with reduced Alpha band power (SHAP contribution = −0.43), indicating compromised relaxation-related brain rhythms.” This level of detail helps clinicians understand the specific EEG biomarkers driving the AI decision.

Validation of Clinical Decisions: In cases where diagnostic uncertainty exists, SHAP explanations serve as a “second opinion” tool. When entropy dominates the feature contributions (>0.6) with negative Alpha rhythm contributions (<−0.3), this pattern reinforces an FTD diagnosis over AD. The system effectively communicates: “Attention, these specific brain activity patterns point more toward FTD than Alzheimer’s. Please verify carefully before finalizing your diagnosis.” This helps clinicians avoid misclassification and increases diagnostic confidence.

Patient and Family Communication: SHAP visualizations enable clinicians to explain AI-driven diagnoses to patients and families in accessible terms: “The brain complexity measures (entropy) in your relative show characteristic patterns of Alzheimer’s disease, visible in these graphs, which explains why our AI system oriented toward this diagnosis.” This transparency builds trust and facilitates informed decision-making.

These applications demonstrate how SHAP transforms the “black box” nature of deep learning into clinically actionable insights, enabling healthcare professionals to understand, validate, and communicate AI-driven diagnostic decisions effectively.

### 5.2. Duration Bias Impact Assessment

Our duration bias mitigation strategies proved effective, as demonstrated by consistent performance metrics across temporal positions within recordings. The segment-based approach successfully normalized contributions across participants with varying recording durations, ensuring that our 98% accuracy reflects genuine disease-related patterns rather than duration-related artifacts.

### 5.3. Class Imbalance Impact Assessment

The effectiveness of our class imbalance mitigation strategies is demonstrated through several key indicators:

Balanced Performance Metrics: Our Bi-LSTM model achieved consistently high performance across all evaluation metrics—precision (99%), recall (99%), and F1-score (99%)—indicating robust performance across all three classes. These balanced metrics suggest that the model is not biased toward the majority class (AD) but performs equally well for the minority class (FTD) and controls.

Clinical Relevance of Learned Features: The SHAP interpretability analysis revealed that entropy emerged as the most discriminative feature, aligning with established neurophysiological literature on neurodegenerative diseases. This consistency suggests that our model learns clinically meaningful patterns rather than spurious correlations that might arise from class imbalance.

Superior Comparative Performance: The substantial improvement over baseline methods (KNN: 38%, SVM: 40%, unidirectional LSTM: 84%) demonstrates that our approach’s superior performance stems from methodological advantages rather than dataset-specific artifacts related to class distribution.

The SHAP summary plot in Figure 8 and Figure 9 demonstrates the relative contributions of key features to the Bi-LSTM model’s predictions. Among these, Entropy emerges as the most impactful feature, significantly influencing the model’s diagnostic decisions. In contrast, the Delta feature exhibits neutrality, contributing neither positively nor negatively to the model’s outputs. This limited effect suggests that Delta does not carry strong discriminative power in this context. These conclusions are derived from experiments conducted on a sample of 100 EEG segments.

In addition to improving model transparency, SHAP feature analysis also holds practical significance in clinical settings. Highlighting features such as Entropy can help clinicians identify critical signal components that reflect pathological brain dynamics. These insights may support clinical interpretation by revealing which aspects of the EEG most influence the model’s decisions, thus enhancing trust in AI-assisted diagnostic tools.

While the results obtained from the segmented EEG data show promising accuracy, it is important to note that this performance pertains specifically to the 5 s segments. EEG signals are inherently long, and we have segmented them into smaller windows, such as 2 s, 5 s, etc., for better processing as we stated before. Ultimately, we chose the 5 s window as it provided optimal results. However, when considering the entire signal, we face a significant challenge: the model still cannot predict the outcomes for the full-length EEG signal at once.

To address this, we explored two potential solutions: Weighted Average of Probabilities and Majority Voting. In the Weighted Average of Probabilities approach, the individual predictions from each segment are aggregated based on their associated probabilities, with more reliable segments (or those with higher probability) contributing more to the final decision. The Majority Voting approach, on the other hand, involves voting for the final classification based on the most frequent prediction across all segments. Both methods help combine the predictions of smaller segments into a single prediction for the full signal.

Despite these approaches, we continue to encounter challenges in achieving robust and accurate predictions when the model is given the entire signal at once. This remains a key limitation in applying our method to whole-signal classification and represents an ongoing hurdle for future work.

## 6. Conclusions and Future Work

This study demonstrates the potential of advanced deep learning models, specifically Autoencoders and Bidirectional LSTM (Bi-LSTM) networks, for classifying EEG signals in the context of neurodegenerative diseases. By effectively reducing the dimensionality of EEG data and capturing crucial temporal patterns, our approach significantly improved the model’s ability to differentiate between Alzheimer’s disease (AD) and frontotemporal dementia (FTD). Additionally, the use of SHapley Additive exPlanations (SHAP) enhanced the interpretability of the model, providing valuable insights into the most influential features for classification, with entropy emerging as a key factor in distinguishing these conditions. Overall, the findings underline the promising application of deep learning for automated and reliable diagnosis of neurodegenerative diseases using EEG signals.

Our future work envisions an ambitious transformation of neurodegenerative disease diagnostics through portable, AI-powered EEG systems that can deliver accurate diagnoses at the point of care. We will prioritize the development of real-time classification models optimized for wearable and portable EEG devices, fundamentally shifting diagnostic paradigms from laboratory-based assessments to accessible, continuous monitoring solutions. This represents a paradigmatic advancement in leveraging EEG’s inherent advantages of portability, non-invasiveness, and cost-effectiveness for widespread clinical deployment.

Building toward comprehensive diagnostic platforms, we will pursue multimodal data integration combining EEG with structural and functional neuroimaging, creating synergistic diagnostic frameworks that capitalize on the complementary strengths of different modalities. Our research scope will expand ambitiously to encompass the full spectrum of neurodegenerative disorders including Parkinson’s disease and Lewy body dementia, ultimately developing universal diagnostic architectures capable of differential diagnosis across multiple conditions.

To establish robust clinical validity, future work will focus on addressing the challenge of classifying full-length EEG signals and validation using larger, multi-center datasets with balanced class distributions. Future studies should include comparison with more recent deep learning architectures such as Transformers, attention-based models, and advanced ensemble methods to provide a more comprehensive performance evaluation. Implementation of advanced resampling techniques such as SMOTE (Synthetic Minority Oversampling Technique) or ADASYN (Adaptive Synthetic Sampling) will provide additional validation, while cross-dataset validation studies will strengthen the generalizability of our methodology beyond current dataset characteristics.

Central to our vision is the seamless translation from research to clinical practice. We will develop clinician-friendly visualization tools and enhance model interpretability to foster clinical trust and adoption. Through strategic collaborations with healthcare institutions and continuous refinement of our methodology, we seek to establish a robust framework that transforms computational research into tangible clinical benefits, ultimately revolutionizing early detection and monitoring of neurodegenerative diseases through accessible, intelligent EEG-based diagnostic systems.

## Figures and Tables

**Figure 1 sensors-25-05690-f001:**
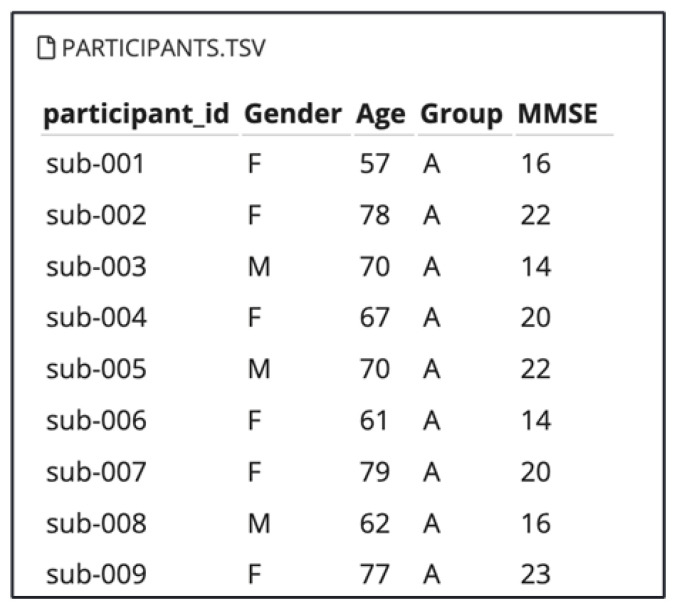
Participants’ information.

**Figure 2 sensors-25-05690-f002:**
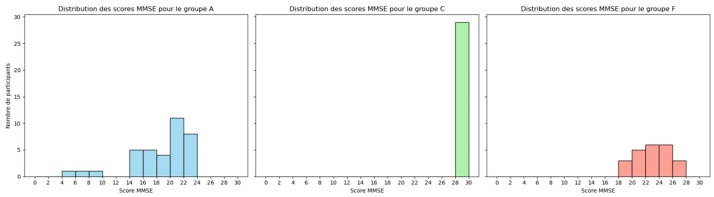
Participants MMSE scores in the tree groups: AD, FTD, and CN.

**Figure 3 sensors-25-05690-f003:**
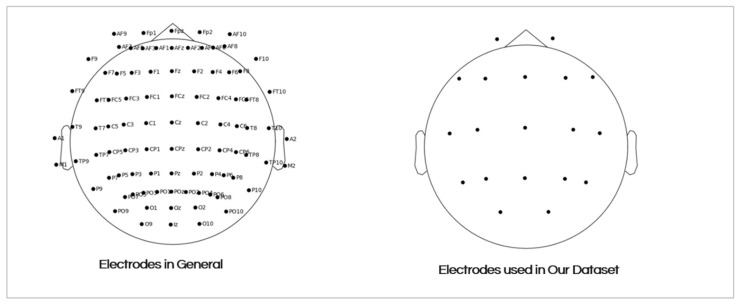
The 19 electrodes’ placements on the scalp.

**Figure 4 sensors-25-05690-f004:**
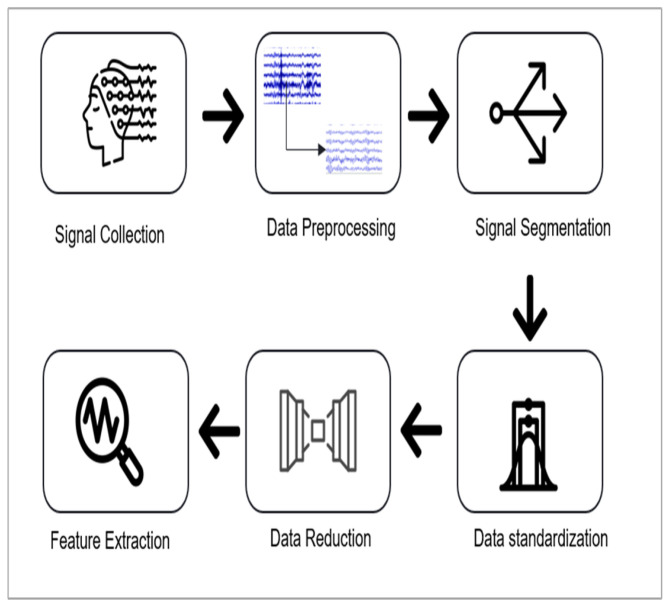
Pre-processing and feature extraction pipeline.

**Figure 5 sensors-25-05690-f005:**
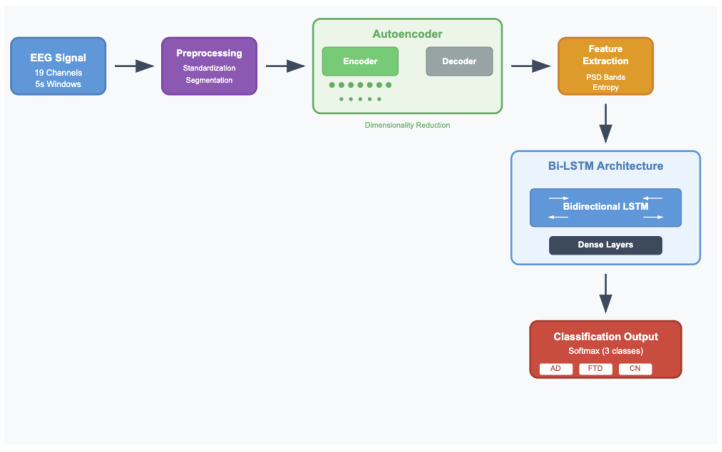
Bi-Directional LSTM EEG classification pipeline with autoencoder and Bi-LSTM architecture.

**Figure 6 sensors-25-05690-f006:**
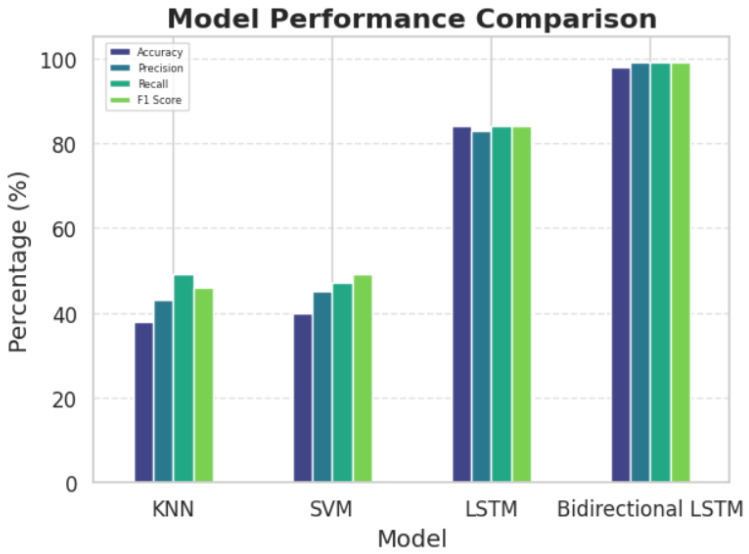
Comparison of classification models performance.

**Figure 7 sensors-25-05690-f007:**
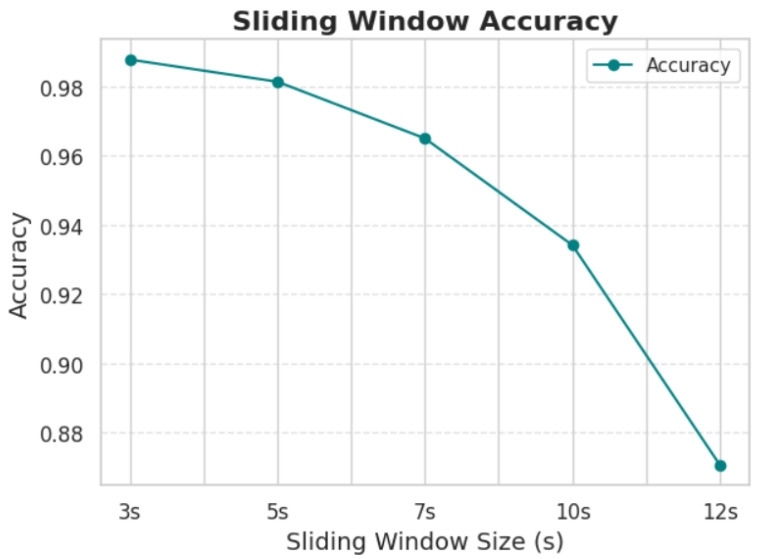
Accuracy comparison of different sliding window lengths.

**Figure 8 sensors-25-05690-f008:**
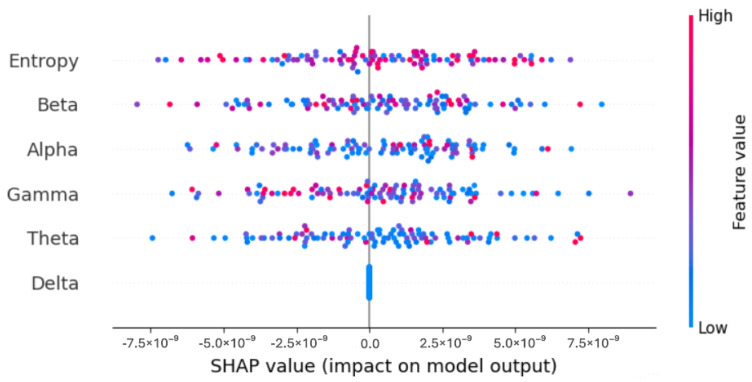
SHAP summary plot.

**Figure 9 sensors-25-05690-f009:**
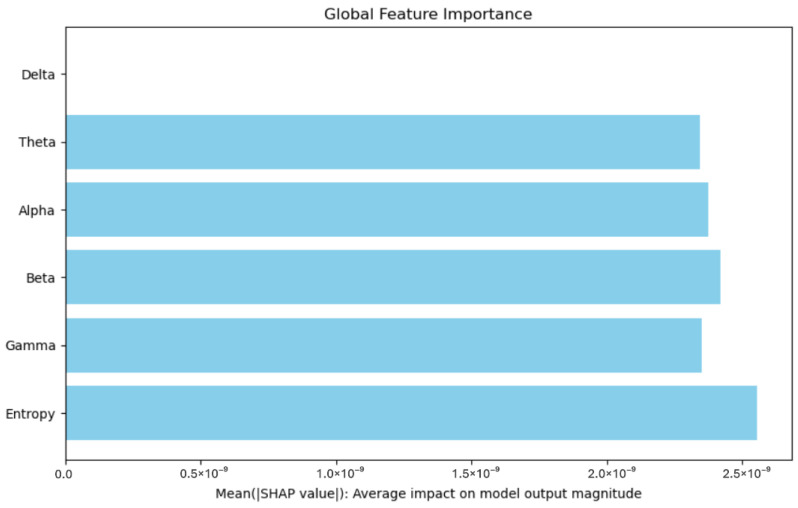
Features importance according to SHAP values.

**Table 1 sensors-25-05690-t001:** Comparison of LSTM and Bidirectional LSTM architectures.

Component	LSTM (Initial Model)	Bi-LSTM (Improved Model)
First Layer	LSTM (64 units, tanh activation)	Bidirectional LSTM (128 units, tanh activation)
Sequence Handling	return_sequences = True	return_sequences = True
Regularization	Dropout (0.3) + Batch Normalization	Dropout (0.3) + Batch Normalization
Second Layer	LSTM (32 units, tanh, return_sequences = False)	LSTM (64 units, tanh, return_sequences = False)
Dense Layers	1 × Dense (32 units, ReLU)	2 × Dense (64 and 32 units, ReLU)
Output Layer	Softmax (3 classes: AD, FTD, CN)	Softmax (3 classes: AD, FTD, CN)
Optimizer	Adam	Adam
Loss Function	Categorical cross-entropy	Categorical crossentropy (with class weights)
Temporal Processing	Unidirectional	Bidirectional (forward + backward)

Note. The Bi-LSTM model incorporates additional improvements, including bidirectional processing and class weight balancing to handle dataset imbalance.

**Table 2 sensors-25-05690-t002:** Performance comparison of classification models.

The Model	Accurancy	Precision	Recall	F1-Score
KNN	38%	43%	49%	46%
SVM	40%	45%	47%	49%
LSTM	84%	83%	84%	71%
Bidrectional LSTM	98%	99%	99%	99%

**Table 3 sensors-25-05690-t003:** Accuracy comparison for different sliding window lengths in EEG signal processing.

Sliding Winodw	3 s	5 s	7 s	10 s	12 s
Accurancy	0.987703	0.981313	0.964953	0.934151	0.879695

## Data Availability

The data presented in this study are openly available in the OpenNeuro repository at https://openneuro.org/datasets/ds004504/versions/1.0.8.

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
