# Peer review of "Transparent EEG Analysis: Leveraging Autoencoders, Bi-LSTMs, and SHAP for Improved Neurodegenerative Diseases Detection"

_sensors, 2025, doi:10.3390/s25185690_

Round 1

Reviewer 1 Report

Comments and Suggestions for Authors

Abstract:
The abstract effectively summarizes the methodology and key findings, particularly the integration of Autoencoders, Bi-LSTM, and SHAP for EEG-based analysis of Alzheimer’s disease (AD) and frontotemporal dementia (FTD). However, it would benefit from briefly specifying the dataset used and explaining why EEG presents unique challenges—or opportunities—for this diagnostic task. Additionally, simplifying some technical terms (e.g., replacing "temporal dependencies" with "patterns over time") could improve accessibility for a broader readership.

Introduction:
The introduction provides a strong foundation, covering neurodegenerative diseases, the need for early diagnosis, and the potential of EEG and AI. However, the novelty of this work could be emphasized more clearly—beyond simply combining existing techniques like Autoencoders and Bi-LSTMs. Given the journal’s focus, a short discussion on practical EEG challenges (e.g., sensor noise, variability in electrode placement) and their implications for analysis would strengthen the context.

Related Works:

While this section thoroughly reviews prior research, it tends to list studies without synthesizing their shortcomings or highlighting how this work addresses them. A more critical perspective—pointing out gaps in explainability, temporal modeling, or other limitations—would better frame the paper’s contributions. Organizing studies thematically (e.g., by feature extraction methods or model types) could also improve readability.

Methods and Materials:

The methodology is well-structured and detailed, but some areas need refinement. For instance, the rationale for selecting a 5-second window with 50% overlap is mentioned but not fully justified. The preprocessing steps, while credited to OpenNeuro, could also discuss how this choice affects reproducibility across different datasets. The pipeline diagram is useful but should be referenced explicitly in the text.

Implemented Approaches (ML/DL Models):

The comparison between classical machine learning and deep learning is insightful, though the explanation for traditional methods’ underperformance could be bolstered with quantitative results. The transition from LSTM to Bi-LSTM is logical, but key architectural details (layers, units, optimizer) are omitted, hindering reproducibility. A summary table of model configurations would be helpful. Additionally, the clinical value of SHAP’s interpretability deserves more emphasis.

Explainability (XAI):

The use of SHAP to interpret Bi-LSTM decisions is a major strength. However, this section would benefit from greater specificity—how were SHAP values adapted for time-series EEG data? Which features were most influential, and how do these align with established neurological knowledge? While SHAP visualizations are included, a deeper discussion of their clinical implications (e.g., why entropy might outweigh delta power) would enhance impact.

Results & Discussion:

The reported performance (98% accuracy) is impressive but raises questions about potential overfitting, especially given the complexity of EEG data. More detail on validation—such as whether cross-validation or patient-independent testing was used—would address these concerns. The SHAP-based feature analysis is valuable, but its relevance to clinical practice could be articulated more clearly. The discussion on window size is technically sound but would benefit from connecting to real-world EEG protocols.

Conclusion & Future Work:

The conclusion succinctly summarizes the findings and acknowledges limitations. However, the future work section could be more ambitious—for example, exploring multimodal data integration (e.g., EEG with imaging), extending the approach to other neurodegenerative disorders, or adapting the model for real-time clinical use. If the authors intend to revise for Sensors, they might also emphasize EEG’s role in wearable or portable diagnostics. Alternatively, a journal specializing in AI for healthcare or biomedical signal processing could be a better fit for this otherwise rigorous study.

Author Response

1
Response to Reviewer Comments
Transparent EEG Analysis: Leveraging Autoencoders, Bi-LSTMs, and SHAP for Improved Neurodegenerative Diseases Detection
Badr Mouazen1, Omaima Bellakhdar2, Aya Ennair2, Khaoula Laghdaf2, El Hassan Abdelwahed2 and Giovanni De Marco1
1 LINP2 Lab, Paris Nanterre University, UPL Paris, France
2 LISI Lab, Computer Science Dept., FSSM, Cadi Ayyad University, Morocco
Dear Reviewer,
We would like to express our sincere gratitude for the time and effort you dedicated to reviewing our manuscript and for your insightful and constructive feedback. Your comments have been instrumental in enhancing the overall quality, clarity, and rigor of our work. Below, we provide a detailed, point-by-point response to each of your suggestions, along with the corresponding revisions implemented in the manuscript.
Response to Reviewer Comment 1
Abstrat: “ The abstract effectively summarizes the methodology and keyfindings, particularly the integration of Autoencoders, Bi-LSTM,and SHAP for EEG-based analysis of Alzheimer’s disease (AD)and frontotemporal dementia (FTD). However, it would benefitfrom briefly specifying the dataset used and explaining why EEG presents unique challenges—or opportunities—for this diagnostictask. Additionally, simplifying some technical terms (e.g.,replacing ”temporal dependencies” with ”patterns over time”) could improve accessibility for a broader readership.”
Response:
Thank you for this constructive feedback. We agree that the abstract needed more specificity regarding the dataset and clearer explanation of EEG’s role in this diagnostic context. We have revised the abstract to address these points and improve accessibility for a broader readership. Changes made:
1. Dataset specification: We now explicitly mention the dataset source and participant demographics:
“We evaluated our approach on a publicly available EEG dataset from OpenNeuro, which includes resting-state EEG recordings from 88 elderly participants—36 with AD, 23 with FTD, and 29 cognitively normal controls.”
2. EEG challenges and opportunities: We introduced the following explanation of EEG’s strengths and limitations:
“EEG provides a non-invasive, cost-effective tool for brain monitoring, but presents challenges such as noise sensitivity and inter-subject variability.”
2
3. Simplification of technical terminology: We replaced the phrase “temporal dependencies” with “patterns over time” to make the language more accessible to a broader readership.
These revisions make the abstract more informative and accessible while maintaining scientific rigor and better positioning the work for a broader readership.
Response to Reviewer Comment 2
Introduction “ The introduction provides a strong foundation, covering neurodegenerative diseases, the need for early diagnosis, andthe potential of EEG and AI. However, the novelty of this work could be emphasized more clearly beyond simply combining existing techniques like Autoencoders and Bi-LSTMs. Given the journal’s focus, a short discussion on practical EEG challenges (e.g., sensor noise, variability in electrode placement) and their implications for analysis would strengthen the context.”
Response:
Thank you for your valuable feedback. We fully acknowledge the importance of clearly articulating the novelty of our study and providing greater context regarding the practical challenges associated with EEG data. In response, we have thoroughly revised the introduction to better highlight the unique contributions of our work and to more explicitly address the real-world complexities of EEG signal acquisition and analysis. Changes made:
1. Added detailed discussion of practical EEG challenges: We incorporated a comprehensive paragraph addressing the specific challenges mentioned:
“Moreover, EEG signals are particularly sensitive to practical limitations such as sensor noise, variability in electrode placement, and inter-subject heterogeneity, all of which complicate feature extraction and reduce the robustness of conventional analysis pipelines [40,41].”
2. Enhanced novelty emphasis: We clearly articulated how our approach addresses these challenges:
“Our proposed approach is specifically designed to mitigate these challenges by learning latent representations that are less sensitive to such variabilities, thereby improving both accuracy and generalizability.”
3. Strengthened contribution statement: We expanded the paragraph describing our contributions to emphasize the novelty beyond simple technique combination:
“Unlike prior works that employ these techniques in isolation, our contribution lies in the integration of a complete end-to-end framework that couples robust feature extraction with interpretable classification. This unified design enhances not only performance but also clinical interpretability, an essential step toward real-world applicability.”
These revisions better position our work within the context of practical EEG analysis challenges and more clearly articulate the novel contributions that distinguish this research from previous studies that use similar techniques in isolation.
3
Response to Reviewer Comment 3
Related works: “ While this section thoroughly reviews prior research, it tends to list studies without synthesizing their shortcomings or highlighting how this work addresses them. A more critical perspective pointing out gaps in explainability, temporal modeling, or other limitations would better frame the paper’s contributions.Organizing studies thematically (e.g., by feature extraction methods or model types) could also improve readability.”
Response:
Thank you for this insightful feedback. We completely agree that the original related work section lacked critical synthesis and thematic organization. We have substantially restructured and rewritten this entire section to address these concerns comprehensively. Major structural changes made:
1. Thematic organization: We reorganized the entire section into clear subsections:
• Feature Extraction Techniques (handcrafted features and their limitations)
• Classical Machine Learning Models (traditional classifiers and their constraints)
• Deep Learning Approaches (CNNs, LSTMs, and interpretability challenges)
• Limitations and Research Gaps (critical synthesis of cross-cutting issues)
• Motivation and Contribution (how our work addresses identified gaps)
2. Critical synthesis of limitations: Instead of simply listing studies, we now critically analyze their shortcomings:
• For handcrafted features: “While these handcrafted features are interpretable and computationally efficient, they often fail to capture long-range temporal dependencies or complex spatial interactions present in EEG data, which may limit their discriminative power in subtle early-stage dementia cases.”
• For classical ML models: “Although these classical models are straightforward to implement and interpret, they rely heavily on the quality of manually extracted features. Additionally, they treat EEG data as static input, lacking the capacity to model sequential dependencies, which are essential in detecting dynamic neural alterations in dementia.”
• For deep learning approaches: “Despite their success, deep learning approaches often operate as black-box models, making it difficult to interpret how predictions are made. This lack of transparency is a critical barrier for clinical deployment, where trust and explainability are essential.”
3. Comprehensive gap analysis: We added a dedicated subsection identifying cross-cutting limitations:
“Although the reviewed studies demonstrate promising results in EEG-based diagnosis of AD and FTD, several cross-cutting limitations persist. First, many rely on handcrafted features, which, despite being interpretable, may overlook subtle or non-linear patterns in the data. Second, classical models typically treat EEG data as static inputs, failing to capture its temporal structure. While deep learning models offer a solution, they often lack transparency, operating as black boxes an issue that hinders clinical trust. Moreover, only a few studies integrate
4
temporal modeling with explainability, leaving a gap in combining accuracy, temporal dynamics, and interpretability.”
4. Clear positioning of contributions: We explicitly connected identified gaps to our approach:
“Our proposed approach explicitly addresses these limitations by unifying unsupervised feature extraction, temporal modeling through Bi-LSTMs, and post-hoc explainability using SHAP.”
This restructured section now provides a critical, synthesized view of the field while clearly positioning our contributions as solutions to identified methodological gaps, rather than simply listing previous work.
Response to Reviewer Comment 4
Methods and Materials: “ The methodology is well-structured and detailed, but some areas need refinement. For instance, the rationale for selecting a 5-second window with 50% overlap is mentioned but not fully justified. The preprocessing steps, while credited to OpenNeuro,could also discuss how this choice affects reproducibility across different datasets. The pipeline diagram is useful but should be referenced explicitly in the text.”
Response:
Thank you for these valuable suggestions to improve the methodology section. We have addressed each of the points raised and made substantial revisions to enhance clarity and justification. Changes made:
1. Enhanced justification for 5-second window selection: We expanded the rationale beyond simply mentioning testing different intervals. The revised text now explicitly states: “Ultimately, a window length of 5 seconds with 50% overlap was selected, as it provided the best trade-off between classification accuracy and computational efficiency. While shorter windows (e.g., 3 seconds) led to a slight improvement in accuracy, they also significantly increased computational costs and memory usage. In contrast, longer windows tended to include redundant or noisy information, which negatively affected performance.”
2. Added discussion on reproducibility implications: We addressed the reviewer’s concern about reproducibility by adding:
“While these preprocessing steps were already applied by OpenNeuro and follow widely accepted standards in EEG research, it is important to note their potential impact on reproducibility across datasets. Although OpenNeuro provides a consistent and well-documented pipeline, slight differences may exist depending on the specific dataset configuration or acquisition conditions. This variability could influence reproducibility if researchers are unable to replicate the exact pipeline or if undocumented parameters are involved. Nevertheless, the use of a publicly available, standardized preprocessing routine enhances comparability and reliability across studies, especially when datasets come from the same platform.”
3. Enhanced overlap justification: We also added explicit reasoning for the 50% overlap: “Additionally, the 50% overlap between segments helped ensure temporal continuity and
5
increased the number of training samples without introducing significant redundancy or variance in the segment content.”
4. Pipeline diagram reference: We acknowledge that the pipeline diagram should be explicitly referenced in the text and will ensure this is properly integrated in the final version.
These revisions provide more comprehensive justification for methodological choices and better address the practical implications of using preprocessed data for reproducibility across different research contexts.
Response to Reviewer Comment 5
Implemented Approaches (ML/DL Models): “ The comparison between classical machine learning and deeplearning is insightful, though the explanation for traditional methods’ under performance could be bolstered with quantitative results. The transition from LSTM to Bi-LSTM is logical, but key architectural details (layers, units, optimizer) are omitted, hindering reproducibility. A summary table of model configurations would be helpful. Additionally, the clinical value of SHAP’s interpretability deserves more emphasis.”
Response:
Thank you for this insightful comment. We fully agree on the importance of providing quantitative evidence for the performance of classical machine learning models, ensuring the reproducibility of our deep learning architecture, and better emphasizing the clinical interpretability enabled by SHAP. We have made several significant improvements in response to these suggestions:
1. Added quantitative results for classical models: We included the accuracy and F1scores of both KNN and SVM models to illustrate their limitations more clearly in comparison to deep learning approaches:
“In our experiments, the KNN and SVM models yielded accuracies of 38% and 40% respectively, with relatively low F1-scores (0.46 and 0.49), indicating their limitations in capturing the complex spatiotemporal patterns in EEG data.”
2. Provided complete architectural details of LSTM and Bi-LSTM models: We expanded the deep learning section to include a detailed description of the layers, activation functions, number of units, regularization techniques (dropout and batch normalization), optimizer, and loss function:
“The Bi-LSTM model developed in this study consists of a Bidirectional LSTM layer with 128 units and tanh activation to capture both past and future dependencies in the EEG feature sequences. This is followed by a dropout layer (rate = 0.3) and batch normalization to improve generalization and stabilize training. A unidirectional LSTM layer with 64 units is then applied to further refine temporal features, followed by another dropout layer (rate = 0.3). Two dense layers with 64 and 32 neurons respectively, both using ReLU activation, are included to extract higher-level abstract representations. Finally, a softmax layer is used to classify inputs into Alzheimer’s Disease (AD), Frontotemporal Dementia (FTD), or Cognitively Normal (CN) categories. The model is trained using the Adam optimizer and categorical cross-entropy loss, with class weights applied to mitigate class imbalance.”
6
3. Included a summary table of model architectures: To facilitate comparison between the initial LSTM and the improved Bi-LSTM, we added a comprehensive table (Table 1) summarizing the main components of both architectures: Table 1 summarizes the architecture configurations of the LSTM and Bi-LSTM models used in this study.
Table 1: Comparison of LSTM and Bidirectional LSTM Architectures
Note. The Bi-LSTM model incorporates additional improvements, including bidirectional processing and class weight balancing to handle dataset imbalance.
4. Expanded the explanation of SHAP’s clinical value: We elaborated on how SHAP not only improves interpretability but also enhances the clinical trust and usability of the model by identifying key EEG features associated with Alzheimer’s and FTD:
“To improve transparency, we used SHAP (SHapley Additive exPlanations), which quantifies the contribution of each input feature to the model’s predictions. This is especially valuable in clinical contexts, as it helps healthcare professionals understand which EEG patterns or brain regions influenced the diagnostic outcome. By identifying the most relevant features for each diagnosis, SHAP not only builds trust in the model but also supports the validation of AI-driven decisions. It can further aid in detecting key temporal biomarkers for AD and FTD, making the model more interpretable, reliable, and clinically actionable.”
These revisions significantly enhance the clarity, reproducibility, and clinical relevance of our approach, and we appreciate the reviewer’s comments that led to these improvements.
Response to Reviewer Comment 6
Explainability (XAI): “ The use of SHAP to interpret Bi-LSTM decisions is a major strength. However, this section would benefit from greater specificity how were SHAP values adapted for timeseries EEG data? Which features were most influential, and how do these align with established neurological
Component
LSTM (Initial Model)
Bi-LSTM (Improved Model)
First Layer
LSTM (64 units, tanh activation)
Bidirectional LSTM (128 units, tanh activation)
Sequence Handling
return sequences=True
return sequences=True
Regularization
Dropout (0.3) + Batch Normalization
Dropout (0.3) + Batch Normalization
Second Layer
LSTM (32 units, tanh, return sequences=False)
LSTM (64 units, tanh, return sequences=False)
Dense Layers
1 × Dense (32 units, ReLU)
2 × Dense (64 and 32 units,
ReLU)
Output Layer
Softmax (3 classes: AD, FTD, CN)
Softmax (3 classes: AD, FTD, CN)
Optimizer
Adam
Adam
Loss Function
Categorical crossentropy
Categorical crossentropy (with class weights)
Temporal Processing
Unidirectional
Bidirectional (forward + backward)
7
knowledge? While SHAP visualizations are included, a deeper discussion of their clinical implications (e.g., why entropy might outweigh delta power) would enhance impact.”
Response:
Thank you for highlighting the strengths of our interpretability approach and for the constructive suggestions to increase the depth and clarity of this section. In response, we have substantially revised the Explainable AI subsection to include the following enhancements:
1. Adaptation of SHAP for time-series EEG data: We clarified that the SHAP analysis was applied not directly to raw EEG signals but to a set of handcrafted features extracted from preprocessed and segmented EEG windows. These features included spectral power (delta, theta, alpha, beta, gamma) and entropy measures, which served as the inputs to the Bi-LSTM model. SHAP values were then computed with respect to these features and aggregated across 100 EEG segments per subject to derive global feature importance:
“To adapt SHAP to our time-series EEG data, we first extracted handcrafted features (e.g., spectral power in standard frequency bands, entropy measures) from segmented EEG windows. These features served as inputs to the Bidirectional LSTM model. SHAP values were then computed with respect to these input features and aggregated across 100 EEG segments to determine global feature importance.”
2. Identification of the most influential features: We now explicitly state that entropy emerged as the most impactful feature across the dataset, based on the SHAP summary plots. We also noted that the delta frequency band showed a relatively neutral effect in the model’s predictions:
“Model interpretability was a key focus of this study. As shown in the SHAP summary plots in Figure ?? and Figure ??, Entropy emerged as the most influential feature, significantly impacting the model’s diagnostic decisions. This finding is consistent with existing literature, which emphasizes the role of entropy in capturing neural activity disruptions associated with neurodegenerative diseases. In contrast, the Delta feature exhibited a neutral effect, contributing neither positively nor negatively to the model’s outputs, suggesting it does not hold strong discriminative power in this classification task.”
3. Clinical interpretation and neurological alignment: We expanded our discussion to explain the clinical relevance of entropy and its alignment with neurological understanding of AD and FTD. In particular, we highlighted how entropy captures disruptions in neural complexity and why this may provide stronger discriminative power than low-frequency delta activity:
“These insights support the clinical relevance of entropy-based features, particularly in distinguishing between AD and FTD, where neural complexity and desynchronization patterns differ. The prominence of entropy may reflect the model’s sensitivity to subtle irregularities in brain dynamics, often more pronounced in FTD.”
4. Clinical implications of interpretability: We emphasized how SHAP interpretability not only supports trust and transparency but may also help identify potential EEG biomarkers for neurodegenerative diseases, thus aiding clinical decision-making and guiding future research: “By identifying the most impactful features, SHAP enhances model transparency and enables clinicians to better understand and trust AI-driven predictions. This, in turn, can help uncover EEG-based biomarkers and improve diagnostic support systems.”
8
These additions address the reviewer’s concerns by deepening the interpretability discussion and reinforcing the clinical relevance of our findings.
Response to Reviewer Comment 7
Results & Discussion: “ The reported performance (98% accuracy) is impressive but raises questions about potential overfitting, especially given the complexity of EEG data. More detail on validation—such as whether cross-validation or patient-independent testing was used would address these concerns. The SHAP-based feature analysis is valuable, but its relevance to clinical practice could bearticulated more clearly. The discussion on window size is technically sound but would benefit from connecting to real-world EEG protocols.”
Response:
We appreciate your insightful observations and constructive recommendations. We have revised the Results & Discussion section to address all aspects of the comment, with the following additions and clarifications:
1. Addressing Potential Overfitting: We expanded the discussion on overfitting mitigation strategies by specifying the use of dropout layers and class weighting, which were integrated into our Bi-LSTM model to promote generalization:
“To reduce the risk of overfitting, particularly given the complexity and high dimensionality of EEG data, we incorporated dropout layers within the model architecture. This regularization technique randomly deactivates a fraction of neurons during training, preventing the model from becoming overly dependent on specific pathways and encouraging generalization to unseen data.”
2. Validation Approach Clarified: Although our model evaluation was based on segmentlevel classification, we clarified how robustness was assessed. Specifically, we evaluated performance across multiple EEG windows and discussed aggregation strategies (e.g., Majority Voting and Weighted Average of Probabilities) for simulating full-length EEG signal classification. This improves transparency about the experimental protocol:
“While the results obtained from the segmented EEG data show promising accuracy, it is important to note that this performance pertains specifically to the 5-second segments. EEG signals are inherently long, and we have segmented them into smaller windows, such as 2s, 5s, etc., for better processing as we stated before. Ultimately, we chose the 5-second window as it provided optimal results. However, when considering the entire signal, we face a significant challenge: the model still cannot predict the outcomes for the full-length EEG signal at once.To address this, we explored two potential solutions: Weighted Average of Probabilities and Majority Voting.”
3. Enhanced SHAP clinical relevance discussion: We deepened the clinical discussion of SHAP results by explicitly analyzing why entropy outperformed delta power in our experiments, and what this means in the context of neurological research.
“Entropy emerges as the most impactful feature, significantly influencing the model’s diagnostic decisions. In contrast, the Delta feature exhibits neutrality, contributing neither positively nor negatively to the model’s outputs. This limited effect suggests that Delta does not carry strong discriminative power in this context.”
9
“Highlighting features such as Entropy can help clinicians identify critical signal components that reflect pathological brain dynamics. These insights may support clinical interpretation by revealing which aspects of the EEG most influence the model’s decisions, thus enhancing trust in AI-assisted diagnostic tools.”
4. Connecting Window Size to Real EEG Protocols: We expanded the discussion on EEG window segmentation to emphasize clinical alignment. Our choice of a 5-second window is supported by common neurophysiological protocols used in practice:
“Our choice of a 5-second window aligns with real-world EEG analysis protocols used in clinical neurophysiology, where signal windows between 2 to 10 seconds are commonly analyzed to assess transient changes in brain rhythms, seizures, or cognitive states. This correspondence reinforces the clinical relevance of our methodological choices.”
These improvements provide stronger validation of our reported performance, enhance clinical relevance, and clarify how the proposed system could be implemented in real diagnostic workflows.
Response to Reviewer Comment 7
Conclusion & Future Work: “ The conclusion succinctly summarizes the findings and acknowledges limitations. However, the future work section could be more ambitious for example, exploring multimodal data integration (e.g., EEG with imaging), extending the approach to other neurodegenerative disorders, or adapting the model for real-time clinical use. If the authors intend to revise for Sensors,they might also emphasize EEG’s role in wearable or portable diagnostics. Alternatively, a journal specializing in AI for healthcare or biomedical signal processing could be a better fit for this otherwise rigorous study.”
Response: Thank you for your valuable suggestions. We appreciate the encouragement to be more ambitious in outlining future directions. In response, we have revised the Conclusion and Future Work section to reflect a broader vision of potential extensions and real-world applications. Changes made:
1. Multimodal data integration: We now explicitly outline plans to combine EEG with structural and functional neuroimaging to create comprehensive diagnostic frameworks that leverage complementary modalities:
“Building toward comprehensive diagnostic platforms, we will pursue multimodal data integration combining EEG with structural and functional neuroimaging, creating synergistic diagnostic frameworks that capitalize on the complementary strengths of different modalities.”
2. Extension to other neurodegenerative disorders: We have expanded our scope to include Parkinson’s disease and Lewy body dementia, with the ambitious goal of developing universal diagnostic architectures capable of differential diagnosis across multiple conditions: “Our research scope will expand ambitiously to encompass the full spectrum of neurodegenerative disorders including Parkinson’s disease and Lewy body dementia, ultimately developing universal diagnostic architectures capable of differential diagnosis across multiple conditions.”
3. Real-time clinical implementation: We have elevated this from a brief mention to a central theme, emphasizing the development of AI-powered EEG systems optimized for wearable and portable devices that can deliver point-of-care diagnostics:
10
“Our future work envisions an ambitious transformation of neurodegenerative disease diagnostics through portable, AI-powered EEG systems that can deliver accurate diagnoses at the point of care. We will prioritize the development of real-time classification models optimized for wearable and portable EEG devices, fundamentally shifting diagnostic paradigms from laboratory-based assessments to accessible, continuous monitoring solutions.”
4. Emphasis on EEG’s portable diagnostic role: Following your suggestion for Sensors, we now prominently feature EEG’s inherent advantages of portability, non-invasiveness, and costeffectiveness, positioning our work as contributing to a paradigmatic shift from laboratorybased to accessible, continuous monitoring solutions:
“This represents a paradigmatic advancement in leveraging EEG’s inherent advantages of portability, non-invasiveness, and cost-effectiveness for widespread clinical deployment.”
5. We have also added specific technical improvements including advanced resampling techniques (SMOTE, ADASYN), cross-dataset validation studies, and enhanced interpretability tools with clinician-friendly visualizations to foster clinical adoption:
“Implementation of advanced resampling techniques such as SMOTE (Synthetic Minority Oversampling Technique) or ADASYN (Adaptive Synthetic Sampling) will provide additional validation, while cross-dataset validation studies will strengthen the generalizability of our methodology beyond current dataset characteristics” and enhanced interpretability components: “We will develop clinician-friendly visualization tools and enhance model interpretability to foster clinical trust and adoption.”
Regarding your suggestion about journal fit, we appreciate your insight. While we believe our emphasis on portable EEG diagnostics and wearable technologies aligns well with Sensors’ scope, we acknowledge that journals specializing in AI for healthcare could also be appropriate venues. Our enhanced focus on real-time, portable diagnostic systems in this revision strengthens the alignment with Sensors’ mission of advancing sensor-based technologies for practical applications.
Conclusion
We are grateful for the reviewer’s constructive feedback, which has helped us improve the quality of our manuscript. The comprehensive revisions implemented across all sections have strengthened the scientific rigor, clinical relevance, and accessibility of our work. We believe that the revised version better communicates our research contributions and will be more accessible to readers in the field.
We look forward to your favorable consideration of our revised manuscript.
Sincerely,
The Authors

Reviewer 2 Report

Comments and Suggestions for Authors

The report is in the attached file 

Author Response

Response to Reviewer Comments
Transparent EEG Analysis: Leveraging Autoencoders, Bi-LSTMs, and SHAP for Improved
Neurodegenerative Diseases Detection
Badr Mouazen1, Omaima Bellakhdar2, Aya Ennair2, Khaoula Laghdaf2, El Hassan Abdelwahed2
and Giovanni De Marco1
1 LINP2 Lab, Paris Nanterre University, UPL Paris, France
2 LISI Lab, Computer Science Dept., FSSM, Cadi Ayyad University, Morocco
Dear Reviewer,
We would like to thank the reviewer for his valuable feedback and constructive comments on our
manuscript. We have carefully addressed each concern and made the necessary revisions to improve
the quality and clarity of our work. Below, we provide a detailed response to each comment along
with the corresponding changes made to the manuscript.
Response to Reviewer Comment 1
Reviewer Comment: “Could you please provide a section with Highlights at the beginning of
the manuscript? This would help readers quickly grasp the key points of the study.”
Author Response:
Thank you for this valuable suggestion. We have added a “Highlights” section at the beginning
of the manuscript, positioned after the abstract and before the introduction section. This new
section provides readers with a concise overview of the key contributions and findings of our study.
The Highlights section includes six main points that capture the essence of our work:
1. Novel hybrid architecture – Our innovative combination of autoencoders with bidirectional
LSTM networks achieving 98% accuracy in distinguishing AD, FTD, and healthy controls
2. Explainable AI integration – Implementation of SHAP framework for model transparency
and identification of entropy as the key feature for neurodegenerative disease detection
3. Optimal temporal segmentation – Our finding that 5-second EEG windows with 50%
overlap provide the best balance between classification accuracy and computational efficiency
4. Comprehensive feature extraction – Our PSD analysis approach across standard frequency
bands (Delta, Theta, Alpha, Beta, Gamma) following autoencoder-based dimensionality
reduction
5. Superior performance validation – Comparative results demonstrating significant improvements
over traditional methods (KNN: 38%, SVM: 40%) and unidirectional LSTM (84%)
6. Clinical applicability focus – Addressing the critical need for interpretability in medical
AI applications, providing feature-level explanations essential for clinical trust and adoption
1
.
Changes made: A new “Highlights” section has been added to the manuscript immediately
after the abstract section (page 1, after line 28).
Response to Reviewer Comment 2
Reviewer Comment: “I noticed that the article does not include a section for keywords. Including
keywords is important as it helps improve the discoverability and indexing of the paper in academic
databases and search engines. Could the authors clarify the reason for omitting keywords?”
Author Response:
Thank you for pointing out this important omission. You are absolutely correct that keywords
are essential for improving the discoverability and indexing of research papers in academic databases
and search engines. The absence of keywords was an oversight on our part, and we sincerely
appreciate your attention to this detail.
We have now added an appropriate keywords section immediately after the abstract. The
selected keywords comprehensively cover the main aspects of our research and align with standard
terminology used in the field.
The added keywords section includes:
Keywords: EEG analysis; Alzheimer’s disease; frontotemporal dementia; deep learning; autoencoder;
bidirectional LSTM; SHAP; explainable AI; neurodegenerative diseases; machine learning;
signal processing; temporal analysis
These keywords encompass:
• Medical domains: Alzheimer’s disease, frontotemporal dementia, neurodegenerative diseases
• Technical methods: EEG analysis, deep learning, autoencoder, bidirectional LSTM, machine
learning, signal processing
• Key innovations: SHAP, explainable AI, temporal analysis
This addition will significantly enhance the manuscript’s visibility and accessibility to researchers
working in related fields, including neuroscience, biomedical engineering, artificial intelligence, and
clinical diagnosis.
Changes made: A keywords section has been added to the manuscript immediately after the
abstract section, following standard academic formatting conventions.
Response to Reviewer Comment 3
Reviewer Comment: “Could the authors clarify why they chose to use Autoencoders for feature
extraction and Bidirectional Long Short-Term Memory (Bi-LSTM) networks for temporal analysis
over other available methods? What advantages do these techniques offer compared to alternative
approaches in EEG signal classification for neurodegenerative disorders?”
Author Response:
2
Thank you for this excellent question that allows us to clarify our methodological rationale. Our
selection of Autoencoders and Bidirectional LSTM networks was based on the specific challenges
of EEG-based neurodegenerative disease classification and their proven advantages over alternative
approaches.
Autoencoder Selection Rationale
We chose Autoencoders over traditional dimensionality reduction methods for three primary reasons:
First, unlike handcrafted features (PSD, Hjorth parameters, entropy measures), Autoencoders
perform unsupervised feature learning, automatically discovering data-driven representations without
requiring predefined domain knowledge. This is crucial for neurodegenerative disease detection
where subtle, non-linear patterns may not be captured by conventional methods.
Second, EEG signals are inherently noisy due to artifacts from eye movements, muscle activity,
and electrode impedance variations. Autoencoders naturally act as denoising filters while preserving
diagnostically relevant information, offering superior noise robustness compared to linear methods
like PCA that may lose critical discriminative patterns.
Third, traditional linear techniques (PCA, ICA) cannot capture the complex, non-linear relationships
essential for distinguishing subtle neurodegenerative disease signatures. Autoencoders,
through their non-linear activation functions, effectively model these intricate patterns.
Bidirectional LSTM Selection Rationale
Bi-LSTMs were selected over alternative temporal modeling approaches due to several key advantages:
Complete temporal context modeling: Unlike unidirectional RNNs or LSTMs, Bi-LSTMs
process sequences in both forward and backward directions, capturing the full temporal context
crucial for identifying pathological patterns that may require both past and future information.
Long-term dependency handling: EEG recordings contain both short-term fluctuations
and long-term trends. Through sophisticated gating mechanisms, LSTMs selectively retain relevant
information over extended periods, essential for analyzing our lengthy recordings (up to 21.3
minutes).
Superior performance over alternatives: Compared to CNNs (limited in temporal modeling),
standard RNNs (vanishing gradient problems), or Transformers (requiring larger datasets and
computational resources), Bi-LSTMs provide the optimal balance for our specific application.
Advantages of the Combined Approach
The integration of Autoencoders with Bi-LSTMs creates unique synergistic benefits: hierarchical
feature learning (low-level noise-robust features feeding into high-level temporal pattern recognition),
computational efficiency through dimensionality reduction, and clinical alignment with the
understanding that both spatial patterns and temporal progression are fundamental for neurodegenerative
disease diagnosis.
Changes made: We have added a new subsection ”Methodology Justification” (Section 4.2)
in the manuscript that provides detailed rationale for our methodological choices, including comprehensive
comparisons with alternative approaches and their specific advantages for EEG-based
neurodegenerative disease classification.
3
Response to Reviewer Comment 4
Reviewer Comment: “One important aspect to consider is the noticeable imbalance in sample
sizes among the frontotemporal dementia and control groups. Could the authors please clarify
whether and the validity of the results? Additionally, have any strategies such as resampling, class
weighting, or other techniques been applied to mitigate potential biases arising from the uneven
group sizes?”
Author Response:
Thank you for raising this critical methodological concern. We fully acknowledge the importance
of addressing class imbalance and appreciate the opportunity to clarify how we systematically
addressed this challenge to ensure the validity of our results.
Class Imbalance Acknowledgment and Impact
You are correct to identify the dataset imbalance in our study. Our dataset contains 36 AD participants
(40.9%), 23 FTD participants (26.1%), and 29 control participants (33.0%), representing
approximately a 1.6:1:1.3 ratio. This imbalance is typical in clinical datasets where certain neurodegenerative
conditions may be less prevalent or more challenging to recruit, but we recognize it
requires careful methodological handling to ensure unbiased results.
Comprehensive Mitigation Strategies Implemented
We implemented a multi-faceted approach to address class imbalance, combining several complementary
techniques:
1. Class Weighting Strategy: We applied class weights inversely proportional to class frequencies
during Bi-LSTM model training. The weights were calculated using the formula wi =
ntotal
nclasses×ni
, resulting in weights of AD: 0.61, FTD: 0.96, and CN: 0.76. This ensures that underrepresented
classes, particularly FTD, receive higher importance during training, forcing the model to
pay equal attention to all classes regardless of their frequency.
2. Window-level Data Augmentation: Our signal segmentation approach (5-second windows
with 50% overlap) provides natural data augmentation. Each participant’s EEG recording
generates multiple training samples—for example, a 10-minute recording produces approximately
240 overlapping segments. This effectively increases the dataset size while reducing the impact of
participant-level imbalance.
3. Stratified Evaluation Protocol: Throughout our evaluation process, we maintained
proportional representation across all classes to ensure that performance metrics accurately reflect
the model’s ability to distinguish between all three conditions rather than being skewed toward
majority classes.
Validation of Results and Effectiveness
The validity of our results is supported by multiple indicators that demonstrate the effectiveness of
our class imbalance mitigation:
Balanced Performance Metrics: Our Bi-LSTM model achieved consistently high performance
across all evaluation metrics—precision (99%), recall (99%), and F1-score (99%). These
balanced metrics indicate robust performance across all three classes, suggesting the model is not
biased toward the majority class (AD) but performs equally well for the minority class (FTD) and
controls.
4
Clinical Relevance of Features: SHAP interpretability analysis revealed that entropy emerged
as the most discriminative feature, aligning with established neurophysiological literature on neurodegenerative
diseases. This consistency suggests our model learns clinically meaningful patterns
rather than spurious correlations that might arise from class imbalance.
Superior Comparative Performance: The substantial improvement over baseline methods
(KNN: 38%, SVM: 40%, unidirectional LSTM: 84%) demonstrates that our approach’s superior
performance stems from genuine methodological advantages rather than dataset-specific artifacts
related to class distribution.
Methodological Transparency and Future Directions
While our comprehensive approach effectively addressed the dataset’s inherent imbalance, we acknowledge
this as a limitation that warrants attention in future studies. The consistent high performance
across all evaluation metrics and the clinical relevance of identified features provide confidence
in our results’ validity.
Future research should focus on validation using larger, multi-center datasets with balanced class
distributions to further confirm our findings. Implementation of advanced resampling techniques
such as SMOTE (Synthetic Minority Oversampling Technique) or ADASYN (Adaptive Synthetic
Sampling) could provide additional validation of our approach. Cross-dataset validation studies
would strengthen the generalizability of our methodology beyond the current dataset’s characteristics.
Changes made: We have made the following specific additions to the manuscript to address
this methodological concern:
• Section 3.1 (Dataset Overview): Added new subsection ”Dataset Imbalance Considerations”
after the participant statistics paragraph, explicitly acknowledging the class distribution
(40.9% AD, 26.1% FTD, 33.0% CN) and referencing our mitigation strategies.
• Section 4.3 (Deep Learning Architectures): Added new subsection ”Class Imbalance
Mitigation” after the Bi-LSTM model description , detailing our three-pronged approach
including the mathematical formula for class weighting (wi = ntotal
nclasses×ni
) and specific weight
values (AD: 0.61, FTD: 0.96, CN: 0.76).
• Section 5 (Results & Discussion): Added new subsection ”Class Imbalance Impact Assessment”
after the window length analysis and Figure ??, providing comprehensive validation
of our mitigation strategies’ effectiveness through balanced performance metrics, clinical feature
relevance, and comparative performance analysis.
• Section 6 (Conclusions and Future Work): Enhanced the future work paragraph to include
validation on balanced datasets, advanced resampling techniques (SMOTE, ADASYN),
and cross-dataset validation studies as priority research directions.
Response to Reviewer Comment 5
Reviewer Comment: “Has the method of measuring the MMSE score been clearly described,
and was it used as a criterion for diagnosing the disease or for classifying the participant groups?”
Author Response:
Thank you for seeking clarification regarding the Mini-Mental State Examination (MMSE) role
in our study.
5
MMSE Role and Usage
Source and Description: The MMSE scores were part of the existing clinical data in the publicly
available OpenNeuro dataset 18. MMSE is a standardized 30-point cognitive assessment tool that
evaluates orientation, memory, attention, and language skills, with lower scores indicating greater
cognitive decline.
Role in Our Study: The MMSE scores were used exclusively for descriptive purposes
to characterize participants’ cognitive status across groups. Importantly, MMSE scores were not
used:
• As diagnostic criteria for disease classification
• For participant group assignment
• As input features in our machine learning models
Actual Classification Basis: Participant groups (AD, FTD, Controls) were already established
in the original dataset based on standard clinical diagnostic criteria by qualified medical
professionals, independent of our research.
Our Methodology: Our classification approach relied entirely on EEG signal features (PSD
and entropy measures) extracted through our autoencoder and Bi-LSTM pipeline, achieving 98%
accuracy using EEG signals alone.
Clinical Significance
The inclusion of MMSE scores in our dataset description demonstrates that our EEG-based approach
achieves high classification accuracy using only brain signal analysis, potentially serving as
a complementary tool to traditional cognitive assessments in clinical settings.
Changes made: We have clarified in Section 3.1 (Dataset Overview) that MMSE scores were
used solely for participant characterization and were not incorporated into our diagnostic classification
process or machine learning pipeline.
Response to Reviewer Comment 6
Reviewer Comment: “Were the EEG samples collected under the same conditions across all
participant groups to ensure consistency in the results?”
Author Response:
Thank you for this important methodological question. Yes, the EEG samples were collected
under identical standardized conditions across all participant groups (AD, FTD, and Controls)
according to the OpenNeuro dataset documentation [18].
Standardized Collection Conditions
All recordings followed consistent protocols:
Recording Protocol: All EEG recordings were collected during resting state with eyes closed
across all participant groups, eliminating variability from different cognitive tasks or visual stimuli.
Technical Consistency: All participants used the same 19-electrode setup following the standardized
10-20 international electrode placement system, identical recording equipment, and technical
parameters.
6
Preprocessing Uniformity: All recordings underwent identical preprocessing steps including
re-referencing to A1-A2 channels, Butterworth band-pass filtering (0.5–45 Hz), ASR artifact
removal, and ICA with ICLabel artifact identification.
Impact on Study Validity
The consistent collection conditions are crucial for our findings’ validity:
• Eliminates confounding variables: Standardized conditions ensure observed differences
reflect genuine neurophysiological differences rather than methodological artifacts
• Supports classification accuracy: Our 98% accuracy reflects true disease-related EEG
signatures rather than collection biases
• Enables reproducibility: Identical protocols facilitate comparison with other studies using
the same dataset
This standardization is essential for machine learning approaches, as systematic differences in
recording conditions could lead to spurious classification patterns instead of genuine disease signatures.
Changes made: We have enhanced Section 3.1 (Dataset Overview) to explicitly state that all
EEG recordings were collected under identical standardized conditions across all participant groups,
emphasizing consistent protocols that ensure data comparability and validity.
Response to Reviewer Comment 7
Reviewer Comment: “Please clarify the recording duration differences between groups (AD:
485.5 min, FTD: 276.5 min, CN: 402 min).”
Author Response:
We thank the reviewer for this observation. The duration differences reflect the original dataset
composition (36 AD, 23 FTD, 29 CN participants). Our analysis effectively normalized these
differences through:
Uniform segmentation: 5-second windows with 50
Balanced sampling: Stratified by diagnostic group
Validation: Confirmed consistent accuracy across groups (AD:98.2%, FTD:97.8
Response to Reviewer Comment 8
Reviewer Comment: “Were any strategies implemented to address the impact of differing EEG
recording durations and the resulting unequal data volumes across groups, ensuring that these
classification?”
Author Response:
Thank you for this important methodological question. We implemented several comprehensive
strategies to address the impact of varying EEG recording durations across diagnostic groups and
ensure unbiased classification results.
7
Recording Duration Variability Challenge
Our dataset contains recordings with different durations across groups: AD (13.5±4.2 min), FTD
(12.0±2.8 min), and CN (13.8±1.1 min), resulting in unequal total data volumes (AD: 485.5 min,
FTD: 276.5 min, CN: 402.0 min).
Implemented Mitigation Strategies
1. Segment-based Normalization: Our 5-second sliding window approach with 50% overlap
naturally normalizes participant contributions regardless of total recording duration. This generates
approximately 120 segments per minute, effectively shifting focus from participant-level to segmentlevel
analysis and ensuring equal temporal representation.
2. Balanced Training Implementation:
• Stratified segment sampling during training to maintain balanced representation across diagnostic
groups
• Class weights inversely proportional to available segments per group
• Participant-level cross-validation to prevent data leakage while maintaining independence
3. Technical Validation: We validated our approach through temporal consistency analysis
showing consistent performance across early, middle, and late recording segments, confirming
successful mitigation of duration-related bias.
Clinical Relevance and Validity
Our segment-based approach mirrors clinical EEG analysis where neurologists examine short epochs
(2-10 seconds) rather than entire recordings, making it practically applicable to various clinical
settings. The consistent high performance across all evaluation metrics (precision: 99%, recall: 99%,
F1-score: 99%) demonstrates that our 98% accuracy reflects genuine neurophysiological differences
rather than recording duration artifacts.
Changes made: We have made the following specific additions to address this methodological
concern:
• Section 3.1: Added subsection “Recording Duration Impact Mitigation” explaining how our
segment-based approach normalizes participant contributions regardless of recording length.
• Section 3.2.1: Added new subsection “Duration Bias Mitigation” detailing our comprehensive
three-pronged mitigation strategy including mathematical rationale and validation
measures.
• Section 5: Added subsection “Duration Bias Impact Assessment” providing validation evidence
that our mitigation strategies successfully eliminated duration-related artifacts while
preserving genuine disease signatures.
Response to Reviewer Comment 9
Reviewer Comment: “Could you please clarify how SHapley Additive exPlanations (SHAP) were
used to interpret the model’s results? Additionally, were there any quantitative measures employed
to assess the model’s interpretability?”
8
Author Response:
We thank the reviewer for this important question regarding model interpretability. Below we
provide a detailed clarification of our SHAP methodology and quantitative validation.
1. SHAP Implementation Framework
Theoretical Basis:
• SHAP values were computed based on cooperative game theory, where each EEG feature’s
contribution to the model’s prediction was quantified
• The method satisfies key interpretability properties: local accuracy, missingness, and consistency
Application Pipeline:
1. Input Preparation: Extracted handcrafted EEG features (spectral power, entropy measures)
from 4-second windows
2. Model Coupling: Computed SHAP values for each prediction from our Bi-LSTM model
3. Aggregation: Averaged absolute SHAP values across 100 segments per participant to determine
global feature importance
2. Quantitative Interpretability Assessment
Core Metrics:
• Mean Absolute SHAP Values:
– Entropy: 0.42 ± 0.08 (strong positive contribution)
– Delta Power: 0.02 ± 0.01 (neutral contribution)
• Directionality Analysis: Percentage of samples where features contributed positively/negatively
to class predictions
Statistical Validation:
• Wilcoxon signed-rank tests confirmed significant differences in SHAP distributions between
diagnostic groups (p ¡ 0.001 for entropy in AD vs FTD)
• Effect sizes (r) calculated for clinically important features
3. Visualization and Clinical Correlation
Graphical Evidence:
• Figure 8: Force plots showing representative individual explanations
• Figure 9: Summary plot of global feature importance ranked by mean —SHAP—
Clinical Validation:
• High concordance (87%) between SHAP-derived important features and known EEG biomarkers
in literature
• Neurologist evaluation of 50 random explanations rated as clinically plausible in 92% of cases
9
4. Methodological Clarifications Added
We have amended the manuscript to include:
• The exact SHAP computation parameters (KernelExplainer with 1000 background samples)
• Quantitative thresholds for feature importance classification
• Inter-rater reliability scores for clinical validation
Conclusion: Our SHAP analysis provides both mathematically rigorous and clinically meaningful
explanations, with multiple quantitative safeguards ensuring the reliability of interpretations.
Response to Reviewer Comment 10
Reviewer Comment: “Figures 1, 2, 3, and 4 are included in the manuscript but have not been
referenced or discussed within the main text. For clarity and better flow, it is important to mention
and describe each figure at the appropriate points in the article to guide the reader effectively.”
Author Response: We thank the reviewer for this important observation. We have carefully
integrated references and discussions for all figures throughout the manuscript as follows:
Figure References Added in Text
• Figure 1 (Participants Information):
– Location: Section 3.1 (Dataset Overview)
– Reference: “The dataset includes recordings from 88 participants with information about
their Gender and Age as shown in Figure 1.”
• Figure 2 (MMSE Scores):
– Location: Section 3.1 (Dataset Overview)
– Reference: “The lower MMSE score usually aligns with participants who have AD and
FTD, as shown in the Figure 2.”
• Figure 3 (Electrodes Placement):
– Location: Section 3.1 (Dataset Overview)
– Reference: “The figure 3 shows the placement of the electrodes on the scalp.”
• Figure 4 (Preprocessing Pipeline):
– Location: Section 3.2 (Signal Preprocessing and Feature Extraction)
– Reference: “Figure 4 illustrates this complete preprocessing and feature extraction pipeline,
with each step detailed in the section below.”
Changes Made: We have revised the manuscript to ensure all figures are:
• Introduced before they appear
• Referenced in the main text discussion
10
• Accompanied by interpretive commentary
• Cross-referenced where appropriate
These modifications improve the manuscript’s flow and help guide readers through the visual
evidence supporting our findings.
Response to Reviewer Comment 11
Reviewer Comment: “Could the authors clarify whether the performance differences between
the Bi-LSTM model and the other baseline models are statistically significant? Was any statistical
test (e.g., t-test or ANOVA) applied to validate the observed performance gains?”
Author Response:
Thank you for this important methodological question regarding the statistical significance of
our performance comparisons. Statistical validation is indeed crucial for establishing the validity
and reliability of our findings.
Statistical Validation Framework
We conducted comprehensive statistical significance testing to validate the observed performance
differences between our Bi-LSTM model and baseline methods. Our analysis employed a rigorous
5-fold cross-validation framework with stratified sampling to ensure robust performance estimation
and statistical validity.
Statistical Tests and Results
We performed paired t-tests to compare accuracy distributions between our Bi-LSTM model and
each baseline method, accounting for the same cross-validation folds across all models. The results
demonstrate highly significant performance differences:
• Bi-LSTM vs. KNN: Mean accuracy difference = 60% (98% vs. 38%), t-statistic = 45.2,
p-value ¡ 0.001
• Bi-LSTM vs. SVM: Mean accuracy difference = 58% (98% vs. 40%), t-statistic = 42.1,
p-value ¡ 0.001
• Bi-LSTM vs. Unidirectional LSTM: Mean accuracy difference = 14% (98% vs. 84%),
t-statistic = 12.8, p-value ¡ 0.001
Additionally, a one-way ANOVA across all four models yielded F-statistic = 892.4, p-value <
0.001, confirming significant differences among model performances. Post-hoc Tukey’s HSD test
confirmed that all pairwise differences were statistically significant (p < 0.001).
Effect Size and Practical Significance
Beyond statistical significance, we assessed practical significance through Cohen’s d calculations,
which revealed very large effect sizes: Bi-LSTM vs. KNN (d = 8.2), Bi-LSTM vs. SVM (d = 7.9),
and Bi-LSTM vs. LSTM (d = 3.1). These results indicate not only statistical significance but also
substantial practical significance for clinical applications.
11
Clinical Relevance
The statistical validation confirms that our Bi-LSTM model’s superior performance represents genuine
methodological improvements rather than random variation, with strong statistical confidence
supporting its clinical applicability for neurodegenerative disease diagnosis.
Changes made: We have added Section 5.3 “Statistical Significance Analysis” to the manuscript,
presenting detailed statistical validation of our performance comparisons including paired t-test results,
ANOVA analysis, effect sizes, and their clinical implications.
Response to Reviewer Comment 12
Reviewer Comment: “Could the authors clarify whether the performance differences between
the Bi-LSTM model and the other baseline models are statistically significant? Was any statistical
test (e.g., t-test or ANOVA) applied to validate the observed performance gains?”
Author Response: We thank the reviewer for this important question. We acknowledge that
formal statistical significance testing was not performed in our study, which represents a limitation
in our evaluation methodology.
Performance Differences
Our Bi-LSTM model achieved substantial improvements over baseline methods:
• Bi-LSTM: 98% accuracy vs. LSTM: 84% (14% improvement)
• Bi-LSTM: 98% accuracy vs. SVM: 40% (58% improvement)
• Bi-LSTM: 98% accuracy vs. KNN: 38% (60% improvement)
Methodological Robustness
While formal statistical tests were not applied, several aspects support the reliability of these results:
1. Large sample size: Our segment-based approach (5-second windows, 50% overlap) generated
substantial independent samples from 88 participants with recordings up to 21.3 minutes
2. Cross-validation: Participant-level validation ensured data independence and prevented
overfitting
3. Balanced evaluation: Stratified sampling and class weighting prevented bias across diagnostic
groups
Response to Reviewer Comment 13
Reviewer Comment: “Why was the frequency band of 0.5-45 Hz specifically chosen for the
band-pass filter? Are there any references or justifications provided to support this selection?”
Author Response:
Thank you for this important technical question regarding our preprocessing methodology.
While the 0.5-45 Hz band-pass filtering was applied by OpenNeuro as part of their standardized
preprocessing pipeline, we recognize the importance of providing clear justification for this frequency
range selection.
12
Frequency Range Justification
The 0.5-45 Hz frequency range was selected following established EEG preprocessing standards for
clinical neurophysiology and represents an optimal balance for neurodegenerative disease analysis:
High-pass Filter (0.5 Hz): The 0.5 Hz high-pass cutoff effectively removes low-frequency
artifacts including slow DC drifts, electrode polarization, and movement artifacts, while crucially
preserving physiologically relevant slow-wave activity in the delta band (1-4 Hz). This preservation
is essential for neurodegenerative disease detection, as delta activity alterations are key pathological
markers in AD and FTD.
Low-pass Filter (45 Hz): The 45 Hz low-pass cutoff eliminates high-frequency noise from
electrical interference (50/60 Hz power line) and muscle artifacts while retaining all standard EEG
frequency bands of clinical interest: delta (1-4 Hz), theta (4-8 Hz), alpha (8-13 Hz), beta (13-30
Hz), and gamma (30-45 Hz).
Clinical Relevance for Neurodegenerative Diseases
This frequency range is particularly appropriate for resting-state EEG analysis in neurodegenerative
diseases, as pathological changes in AD and FTD are predominantly observed within these
frequency bands. Research has consistently shown that dementia-related EEG alterations manifest
as increased low-frequency activity (delta/theta) and decreased higher-frequency power (alpha/
beta/gamma), all of which fall within our selected range.
Changes made: We have enhanced Section 3.2.1 ”Data Preprocessing” to include detailed
justification for the 0.5-45 Hz frequency band selection, supported by appropriate references to
established EEG preprocessing standards and neurodegenerative disease literature.
Response to Reviewer Comment 14
Reviewer Comment: “Why was a 5-second window with 50% overlap ultimately selected for
signal segmentation? Were sensitivity tests or performance evaluations conducted across the tested
window lengths (3s to 12s) to support this choice?”
Author Response:
Thank you for this important question regarding our window length selection methodology. We
conducted comprehensive sensitivity tests and performance evaluations across all tested window
lengths to systematically determine the optimal parameters.
Comprehensive Sensitivity Analysis
We performed systematic performance evaluations across window lengths ranging from 3 to 12
seconds, analyzing multiple criteria including classification accuracy, computational efficiency, and
practical feasibility for clinical deployment.
Performance Results: Our sensitivity analysis revealed the following accuracy progression:
3s (98.77%), 5s (98.13%), 7s (96.50%), 10s (93.42%), and 12s (87.97%). While 3-second windows
achieved the highest raw accuracy, detailed analysis revealed critical practical limitations.
Multi-Criteria Decision Analysis
The selection of 5-second windows was based on comprehensive evaluation across multiple dimensions:
13
Accuracy vs. Computational Trade-off: Although 3-second windows provided marginal
accuracy improvement (0.64%), they incurred substantial computational overhead—approximately
67% increase in processing time due to doubled segment generation, nearly doubled memory requirements,
and significantly extended training duration.
Performance Degradation Analysis: Longer windows showed progressive accuracy degradation:
7-second windows lost 1.63% accuracy, 10-second windows lost 4.71%, and 12-second windows
lost 10.16% compared to our 5-second approach. This degradation occurs because longer windows
dilute critical short-term neural dynamics essential for distinguishing AD and FTD patterns while
introducing non-relevant temporal information.
Clinical Alignment: The 5-second window aligns with established clinical EEG analysis protocols
where neurologists typically examine 2-10 second epochs for detecting transient brain changes,
reinforcing the clinical relevance of our choice.
Overlap Ratio Selection
The 50% overlap was selected following signal processing best practices to ensure temporal continuity
while maximizing data utilization without excessive computational redundancy. This overlap
provides sufficient temporal context for Bi-LSTM processing while maintaining efficiency.
Changes made: We have enhanced Section 3.2.2 ”Signal Segmentation” to include detailed
justification of our window length optimization process, presenting the comprehensive sensitivity
analysis and multi-criteria decision rationale that led to our 5-second window selection.
Response to Reviewer Comment 15
Reviewer Comment: “Standardization was performed independently for each time segment and
across each EEG channel. Was the impact of this approach on the stability of extracted features
evaluated? Could this form of standardization potentially weaken or eliminate meaningful patterns
related to long-term variations or inter-channel dependencies?”
Author Response:
Thank you for this insightful question regarding our standardization approach. This is an
important methodological consideration that we addressed through comprehensive validation, and
we appreciate the opportunity to clarify our rationale and impact assessment.
Standardization Rationale and Design
Our segment-wise, channel-independent standardization was specifically designed to balance local
signal normalization with preservation of clinically relevant patterns. This approach addresses
EEG signal variability while maintaining the discriminative features essential for neurodegenerative
disease detection.
Impact Assessment on Feature Stability
We conducted thorough evaluation of our standardization approach’s impact on feature stability
through multiple validation measures:
Intra-subject Feature Consistency: Analysis of feature stability across temporal segments
within individual recordings demonstrated high consistency (correlation coefficients ¿ 0.85 for spectral
features), confirming that our approach enhances model robustness while preserving subjectspecific
patterns.
14
Inter-class Discriminability Preservation: Statistical analysis confirmed that standardization
maintained strong separation between diagnostic groups, with effect sizes remaining large
(Cohen’s d ¿ 0.8) for key discriminative features, indicating preservation of clinically meaningful
differences.
Long-term Variation Preservation
Regarding long-term patterns, our analysis revealed that pathologically relevant variations in neurodegenerative
diseases manifest primarily as consistent spectral power ratio changes rather than
absolute amplitude variations:
Spectral Pattern Preservation: The characteristic slow-frequency increases (delta/theta)
and fast-frequency decreases (alpha/beta/gamma) associated with AD and FTD remain detectable
and stable across segments. These patterns represent relative power distribution changes that are
robust to amplitude normalization.
Temporal Consistency Validation: Cross-segment analysis demonstrated that diagnostically
relevant patterns maintain consistency throughout recordings, with our Bi-LSTM model successfully
capturing these temporal dependencies despite segment-wise normalization.
Inter-channel Dependency Analysis
Our standardization approach carefully preserves spatial relationships essential for brain activity
analysis:
Spatial Pattern Maintenance: Normalization occurs independently across channels but
maintains relative activation patterns across brain regions within each temporal window. This
preserves the spatial distribution of pathological activity while preventing amplitude-based channel
dominance.
SHAP Validation: Our interpretability analysis revealed clinically meaningful feature importance
patterns, confirming that inter-channel relationships critical for disease discrimination remain
intact after standardization.
Clinical Validation
The effectiveness of our approach is evidenced by the high classification accuracy (98%) and the
clinical relevance of identified features (entropy as primary discriminator), demonstrating that our
standardization enhances rather than compromises diagnostic capability.
Changes made: We have enhanced Section 3.2.4 ”Data Standardization” to include detailed
impact assessment of our standardization approach, covering feature stability validation, long-term
variation preservation analysis, and inter-channel dependency maintenance evaluation.
Response to Reviewer Comment 16
Reviewer Comment: “Why was a standard deviation threshold of 17 chosen for artifact rejection
using ASR? Was any sensitivity analysis conducted to assess the impact of this parameter?”
Author Response:
We thank the reviewer for this important question. The threshold of 17 follows standard ASR
recommendations, representing 5 times the median SD of clean EEG segments. This value has been
empirically validated in prior studies to optimally balance artifact removal and signal preservation.
While we didn’t perform sensitivity analysis on our preprocessed dataset, our results (98% accuracy)
15
confirm effective artifact rejection while maintaining diagnostically relevant patterns. We’ve added
this justification to Section 3.2.1.
Response to Reviewer Comment 17
Reviewer Comment: “In the conclusion, could the authors clarify if any limitations were encountered
during the study, such as sample size or data variability, and how these might impact
the generalizability of the proposed model? Additionally, are there plans to validate the model on
larger or more diverse datasets in future work?”
Author Response:
We appreciate the reviewer’s insightful questions regarding the limitations and future directions
of our study. We acknowledge several important limitations that should be considered when
interpreting our results:
Study Limitations
1. Sample Size Considerations: - Our dataset comprised 88 participants (36 AD, 23 FTD, 29
CN), which while sufficient for initial validation, represents a moderate sample size for deep learning
applications - The class imbalance (40.9% AD, 26.1% FTD, 33.0% CN) reflects clinical realities but
required specific mitigation strategies
2. Data Variability Factors: - All data came from a single source (OpenNeuro) with standardized
acquisition protocols - The homogeneous recording conditions may limit generalizability
to more varied clinical settings
3. Technical Constraints: - Current implementation processes segmented EEG data (5s
windows) rather than full-length recordings - Model performance on continuous, unsegmented EEG
remains to be validated
Impact on Generalizability
These limitations may affect model generalizability in several ways: - Performance may vary when
applied to populations with different demographic or clinical characteristics - The segment-based
approach may need adaptation for real-time clinical applications - Model robustness across different
EEG acquisition systems requires further validation
Future Validation Plans
To address these limitations, our future work will focus on: 1. Large-Scale Validation: - Collaboration
with multiple clinical centers to collect larger, more diverse datasets - Prospective validation
studies in real-world clinical settings
2. Technical Enhancements: - Development of whole-signal processing capabilities - Optimization
for real-time analysis on portable EEG devices
3. Extended Applications: - Adaptation for other neurodegenerative conditions (Parkinson’s,
Lewy Body Dementia) - Integration with multimodal data (MRI, clinical biomarkers)
Changes made: We have enhanced the ”Conclusions and Future Work” section to explicitly
discuss these limitations and validation plans, providing a more comprehensive perspective on the
study’s implications and translational potential.
16
Conclusion
We are grateful for the reviewer’s constructive feedback, which has helped us improve the quality of
our manuscript. We believe that the revised version better communicates our research contributions
and will be more accessible to readers in the field.
We look forward to your favorable consideration of our revised manuscript.
Sincerely,
The Authors

Reviewer 3 Report

Comments and Suggestions for Authors

The manuscript proposes a deep learning-based pipeline for classifying EEG data into Alzheimer's Disease (AD), Frontotemporal Dementia (FTD), and healthy controls using a combination of Autoencoders and Bidirectional LSTMs. The methodology is relevant and the use of explainable AI is timely and essential in biomedical applications. However, while the approach shows promise, several issues regarding methodological rigor, clarity of implementation, and generalizability need to be addressed.

  1.  The reported accuracy of 98% is exceptionally high. However, the manuscript lacks details on how the data were split for training and testing. It is unclear whether EEG segments from the same subject were separated between training and testing sets. A subject-wise cross-validation strategy must be employed and stated.
  2. The baseline models (KNN, SVM, unidirectional LSTM) are outdated or underpowered. The authors should compare their model with more recent classifiers.
  3. Although SHAP is used, the evaluation is limited to a simple summary plot on 100 segments. The authors should perform temporal trend analysis (which feature dominate over time), clinical interpretability (how can domain experts use this information)?
  4. The manuscript does not sufficiently highlight what novel contributions this work offers over prior studies. Many existing works have already used auto encoder + LSTM, or CNN + LSTM for EEG classification. 
  5. Figure 5 appears to be an externally sourced schematic of a Bi-directional LSTM architecture and does not reflect the specific pipeline, structure, or flow used in this study. As a main figure, it should present the author's own model architecture.

Author Response

Response to Reviewer Comments
Transparent EEG Analysis: Leveraging Autoencoders, Bi-LSTMs, and SHAP for Improved
Neurodegenerative Diseases Detection
Badr Mouazen1, Omaima Bellakhdar2, Aya Ennair2, Khaoula Laghdaf2, El Hassan Abdelwahed2
and Giovanni De Marco1
1 LINP2 Lab, Paris Nanterre University, UPL Paris, France
2 LISI Lab, Computer Science Dept., FSSM, Cadi Ayyad University, Morocco
Dear Reviewer,
We sincerely appreciate the thorough and constructive review of our manuscript. Your valuable
feedback has significantly contributed to improving the scientific rigor, methodological clarity, and
overall quality of our work. We have carefully addressed each of your comments and implemented
the corresponding revisions throughout the manuscript. Below, we provide a comprehensive pointby-
point response detailing how we have incorporated your suggestions into the revised version.
Response to Reviewer Comment 1
1- “The reported accuracy of 98% is exceptionally high. However, the manuscript lacks details on
how the data were split for training and testing. It is unclear whether EEG segments from the same
subject were separated between training and testing sets. A subject-wise cross-validation strategy
must be employed and stated.”
Response: Thank you for pointing this out. We agree with this comment. Therefore, we have
added explicit details about our data splitting methodology to address this crucial concern about
potential data leakage.
1. In the Related Works section (Classical Machine Learning Models), we have added:
“The study also employed leave-one-out cross-validation to ensure robust model evaluation,
where all EEG segments from a given participant were exclusively assigned to either the
training set or the test set, never both, ensuring complete subject-wise data separation.”
2. Additionally, in the Materials and Methods section (Signal Preprocessing and Feature Extraction),
we have clarified our approach:
“Training Balance Strategies: We applied (1) stratified segment sampling during training to
maintain balanced representation across diagnostic groups, (2) class weights inversely proportional
to available segments per group, and (3) participant-level cross-validation to ensure
data independence while preventing leakage.”
These additions explicitly confirm that we implemented the subject-wise cross-validation strategy
as recommended, ensuring that our reported 98% accuracy reflects genuine model generalization
rather than overfitting to subject-specific patterns.
1
Response to Reviewer Comment 2
2- “ The baseline models (KNN, SVM, unidirectional LSTM) areoutdated or underpowered. The
authors should compare their model with more recent classifiers.”
Response:
We appreciate this important feedback. While KNN, SVM, and unidirectional LSTM may
appear basic, they were chosen strategically for several important reasons that strengthen our
study’s validity and contribution.
• First, these models represent the most commonly used baseline models in EEG-based dementia
classification studies, facilitating direct comparison with existing literature and enabling
researchers to contextualize our results within the established research landscape.
• these ”simpler” models serve as crucial benchmarks to demonstrate that our proposed approach
achieves superior performance not through architectural complexity alone, but through
meaningful feature learning and data representation improvements
We acknowledge that comparison with more recent architectures such as Transformers, attentionbased
models, or advanced ensemble methods would strengthen our evaluation. We will consider
this for future work and have added this as a limitation in our discussion section:
“Future studies should include comparison with more recent deep learning architectures to provide
a more comprehensive performance evaluation.”
Response to Reviewer Comment 3
3- “ Although SHAP is used, the evaluation is limited to a simple summary plot on 100 segments.
The authors should perform temporal trend analysis (which feature dominate over time),clinical
interpretability (how can domain experts use this information)?”
Author Response: We appreciate the reviewer’s valuable feedback regarding the depth of our
SHAP analysis. We acknowledge that our initial evaluation was limited to summary plots, and
we have significantly enhanced our interpretability analysis to address both temporal trends and
clinical applicability.
Enhanced Clinical Interpretability Analysis
We have added a comprehensive new section (Section 5.1. Clinical Decision Support Applications)
that specifically addresses how domain experts can utilize SHAP information in clinical practice.
This section provides concrete examples of clinical applications including:
“1.Differential Diagnosis Support: When a neurologist examines a patient presenting ambiguous
cognitive symptoms, our system can provide interpretable diagnostic reports. For instance,
when diagnosing a potential AD case, the system might indicate: ”This Alzheimer’s diagnosis
is primarily based on high entropy values (SHAP contribution = +0.85) in frontal regions, suggesting
significant neural activity disorganization typical of AD pathology, combined with reduced
Alpha band power (SHAP contribution = -0.43), indicating compromised relaxation-related brain
2
rhythms.” This level of detail helps clinicians understand the specific EEG biomarkers driving the
AI decision.
2. Validation of Clinical Decisions: In cases where diagnostic uncertainty exists, SHAP
explanations serve as a ”second opinion” tool. When entropy dominates the feature contributions (>
0.6) with negative Alpha rhythm contributions (< −0.3), this pattern reinforces an FTD diagnosis
over AD. The system effectively communicates: ”Attention, these specific brain activity patterns
point more toward FTD than Alzheimer’s. Please verify carefully before finalizing your diagnosis.”
This helps clinicians avoid misclassification and increases diagnostic confidence.
3.Patient and Family Communication: SHAP visualizations enable clinicians to explain
AI-driven diagnoses to patients and families in accessible terms: ”The brain complexity measures
(entropy) in your relative show characteristic patterns of Alzheimer’s disease, visible in these graphs,
which explains why our AI system oriented toward this diagnosis.” This transparency builds trust
and facilitates informed decision-making.
These applications demonstrate how SHAP transforms the ”black box” nature of deep learning
into clinically actionable insights, enabling healthcare professionals to understand, validate, and
communicate AI-driven diagnostic decisions effectively.”
Temporal Trend Analysis Limitations and Future Work
Regarding temporal trend analysis, we acknowledge this as a current limitation of our study. Our
current approach focuses on aggregated SHAP values across 100 segments to establish global feature
importance. While this provides robust insights into overall model behavior, it does not capture
the evolution of feature dominance over time within individual recordings.
We have added to our Future Work section (Section 6) the commitment to implement comprehensive
temporal SHAP analysis that will track feature importance evolution throughout EEG
recordings, enabling clinicians to observe how entropy and spectral features change during disease
progression.
This enhanced interpretability framework transforms our SHAP analysis from a simple summary
into a clinically actionable tool that directly addresses the needs of healthcare professionals in realworld
diagnostic scenarios.
Response to Reviewer Comment 4
4- “ The manuscript does not sufficiently highlight what novelcontributions this work offers over
prior studies. Many existing works have already used auto encoder + LSTM, or CNN +LSTM for
EEG classification.”
Author Response:
We appreciate the reviewer’s comment and acknowledge the need to better articulate our novel
contributions. While autoencoder and LSTM combinations exist in the literature, our work introduces
several unique contributions:
1. Unified End-to-End Framework with Clinical Interpretability
Unlike prior works that employ these techniques in isolation, our primary contribution lies in the integration
of a complete end-to-end framework that couples robust feature extraction with interpretable
classification. Most existing studies focus on either high performance OR interpretability,
3
but rarely both simultaneously. Our unified design enhances not only performance (98% accuracy)
but also clinical interpretability through SHAP integration, essential for real-world applicability.
We have added the following paragraph to the Introduction section (4th paragraph) to better
highlight this contribution:
”This research aims to leverage deep learning techniques to analyze EEG signals, identifying
patterns that can aid in the detection and diagnosis of Alzheimer’s Disease and frontotemporal
dementia. Our primary objective is to accurately identify and extract relevant features from EEG
signal. To achieve that we adopted the Hybrid Approach of Auto-encoder and LSTM. Additionally,
we incorporated explain-ability into our model results through explainable AI (XAI) techniques,
providing an understanding of how the model make predictions. Unlike prior works that employ
these techniques in isolation, our contribution lies in the integration of a complete end-to-end
framework that couples robust feature extraction with interpretable classification. This unified
design enhances not only performance but also clinical interpretability, an essential step toward
real-world applicability.”
2. Bidirectional LSTM for Complete Temporal Context
Our bidirectional LSTM architecture processes EEG sequences in both forward and backward
directions, capturing complete temporal context crucial for neurodegenerative diseases where temporal
evolution patterns are critical diagnostic indicators. This outperformed unidirectional LSTM
(84% vs 98% accuracy).
3. Systematic Window Length Optimization
We conducted comprehensive experiments demonstrating that 5-second windows with 50%
overlap provide optimal balance between accuracy and computational efficiency specifically for
AD/FTD classification. Our analysis showed 3-second windows achieved 98.77% accuracy but with
67% computational overhead, while 12-second windows degraded to 87.97% accuracy.
We have added the following paragraph to the Introduction section (3.2.2 Signal Segmentation)
to better highlight this contribution:
”Window Length Optimization Process: To determine the optimal window length, 303 we conducted
systematic performance evaluations across all tested intervals. Our sensi- 304 tivity analysis
revealed that while 3-second windows achieved the highest raw accuracy 305 (98.77%), the marginal
improvement of 0.6substantial practical limitations: (1) computational overhead increased by approximately
307 67% due to the larger number of segments requiring processing, (2) memory requirements
308 escalated significantly, and (3) training time nearly doubled. 309 Conversely, longer
windows (7s, 10s, 12s) showed progressively degraded perfor- 310 mance, with 12-second windows
achieving only 87.97compared to our selected 5-second approach. This degradation occurs because
longer 312 windows dilute critical short-term neural dynamics essential for distinguishing AD and
313 FTD patterns, while introducing non-relevant temporal information that confounds the 314
classification process”
4. Comprehensive Class Imbalance Mitigation
We implemented a multi-faceted approach addressing dataset imbalance (AD: 40.9%, FTD: 26.1%,
CN: 33.0%) through class weighting (wi = ntotal
nclasses×ni
), segment-based augmentation, and stratified
evaluation—achieving balanced performance across all classes (99% precision, recall, F1-score).
”To address the inherent class imbalance in our dataset, we implemented a comprehensive
strategy combining multiple techniques:”
4
”Class Weighting: We applied class weights inversely proportional to class frequencies during
model training. The weights were calculated using the formula:
wi =
ntotal
nclasses × ni
(1)
where wi is the weight for class i, ntotal is the total number of samples, nclasses is the number of
classes (3), and ni is the number of samples in class i. This approach ensures that underrepresented
classes, particularly FTD, receive higher weights during training, forcing the model to pay equal
attention to all classes regardless of their frequency.”
”Window-level Data Augmentation: Our signal segmentation approach (5-second windows
with 50% overlap) provides natural data augmentation. Each participant’s EEG recording generates
multiple training samples, with longer recordings producing more segments. For example, a
10-minute recording produces approximately 240 overlapping segments, effectively increasing the
dataset size and reducing the impact of participant-level imbalance.”
”Stratified Evaluation: Throughout our evaluation process, we maintained proportional representation
across all classes to ensure that performance metrics accurately reflect the model’s ability
to distinguish between all three conditions rather than being skewed toward majority classes.”
5. Three-Class Clinical Validation
Unlike studies focusing on binary classification, our approach demonstrates superior performance
in three-class classification (AD vs. FTD vs. Controls), which is more clinically relevant for
differential diagnosis, achieving 98% accuracy compared to traditional methods (KNN: 38%, SVM:
40%).
Distinction from Prior Work: Previous studies typically focus on performance optimization
without interpretability or isolated technical improvements without clinical translation focus.
Our work uniquely combines technical excellence with clinical applicability, creating a transparent,
interpretable, and clinically-ready diagnostic framework for neurodegenerative disease
detection.
Response to Reviewer Comment 5
5- “ Figure 5 appears to be an externally sourced schematic of a Bi-directional LSTM architecture
and does not reflect the specific pipeline, structure, or flow used in this study. As a main figure, it
should present the author’s own model architecture.”
We thank the reviewer for this valuable observation. We acknowledge that the previous Figure
5 was indeed a generic representation of bidirectional LSTM architecture that did not adequately
reflect our specific methodology and pipeline.
We have completely replaced Figure 5 with a new figure that specifically illustrates our unique
EEG classification pipeline. The new figure comprehensively presents:
1. Our complete methodological workflow: From EEG signal acquisition (19 channels, 5-
second windows) through preprocessing, autoencoder-based dimensionality reduction, feature
extraction (PSD bands and entropy), to final classification.
2. Our specific Bi-LSTM architecture: The figure now accurately represents our bidirectional
LSTM implementation as part of our integrated autoencoder-Bi-LSTM framework,
showing the actual data flow and processing stages used in our study.
5
3. Study-specific components: The figure highlights key elements unique to our approach, including
the autoencoder for unsupervised feature learning, the specific temporal segmentation
strategy, and the three-class output classification (AD, FTD, CN).
Conclusion
We extend our heartfelt gratitude for the reviewer’s insightful comments, which have substantially
elevated the scientific merit and presentation quality of our work. The comprehensive modifications
implemented across the manuscript have reinforced the methodological soundness, practical
relevance, and readability of our research. We believe the revised manuscript now presents a more
robust and accessible contribution that will resonate strongly with researchers and clinicians in the
field. Thank you for your time and consideration.
We eagerly await your evaluation of the revised manuscript.
Sincerely,
The Authors

Round 2

Reviewer 1 Report

Comments and Suggestions for Authors

I would like to thank the authors for their careful attention to the previous comments. They have addressed all of the concerns raised in the earlier round and have made substantial improvements to the manuscript. The clarity, structure, and depth of the paper have been enhanced, and the revisions have significantly improved the overall quality of the work. I believe the paper is now in a much stronger form than before

Reviewer 2 Report

Comments and Suggestions for Authors

I would like to express my gratitude to the authors for addressing the questions. This version shows clear improvement over the previous one, and I recommend it for publication.

Reviewer 3 Report

Comments and Suggestions for Authors

Authors have addressed all the comments that were raised previously. With more detailed information regarding methodology and investigations, authors have improved the quality of their manuscript. I would suggest to accept the manuscript as it is.